# Self-Supervised Learning via Maximum Entropy Coding

**Xin Liu**     **Zhongdao Wang**     **Yali Li**     **Shengjin Wang**[*]

Beijing National Research Center for Information Science and Technology (BNRist)
Department of Electronic Engineering, Tsinghua University
`{xinliu20, wcd17}@mails.tsinghua.edu.cn`
`{liyali13, wgsgj}@tsinghua.edu.cn`

## Abstract

A mainstream type of current self-supervised learning methods pursues a general-purpose representation that can be well transferred to downstream tasks, typically by optimizing on a given pretext task such as instance discrimination. In this work, we argue that existing pretext tasks inevitably introduce biases into the learned representation, which in turn leads to biased transfer performance on various downstream tasks. To cope with this issue, we propose Maximum Entropy Coding (**MEC**), a more principled objective that explicitly optimizes on the structure of the representation, so that the learned representation is less biased and thus generalizes better to unseen downstream tasks. Inspired by the principle of maximum entropy in information theory, we hypothesize that a generalizable representation should be the one that admits the maximum entropy among all plausible representations. To make the objective end-to-end trainable, we propose to leverage the minimal coding length in lossy data coding as a computationally tractable surrogate for the entropy, and further derive a scalable reformulation of the objective that allows fast computation. Extensive experiments demonstrate that MEC learns a more generalizable representation than previous methods based on specific pretext tasks. It achieves state-of-the-art performance *consistently* on various downstream tasks, including not only ImageNet linear probe, but also semi-supervised classification, object detection, instance segmentation, and object tracking. Interestingly, we show that existing batch-wise and feature-wise self-supervised objectives could be seen equivalent to low-order approximations of MEC. Code and pre-trained models are available at `https://github.com/xinliu20/MEC`.

## 1   Introduction

Self-supervised learning (SSL) aims to learn rich and meaningful representations without relying on human annotations. Pursuing *general-purpose* representations, SSL models are typically used as pre-trained weights for providing a good initialization to downstream tasks. In this sense, SSL [12, 31, 10, 29, 15, 14, 86, 30] has seen great progress in computer vision, and can achieve competitive or even better performance on various downstream tasks compared to its supervised counterparts.

At the core of current SSL methods is the design of *pretext tasks*. A pretext task is a (usually hand-crafted) learning objective in which the supervision signals could be mined from the data itself, with the aim of applying the learned representation to other downstream tasks. Early attempts of pretext tasks typically discard a certain property of the input data and then force the model to predict the discarded property. For instance, one can convert an RGB image to a gray-scale one, and train the model to predict the original color [87, 88]; or apply random rotation to image patches and ask the

---

[*]Corresponding author

36th Conference on Neural Information Processing Systems (NeurIPS 2022).

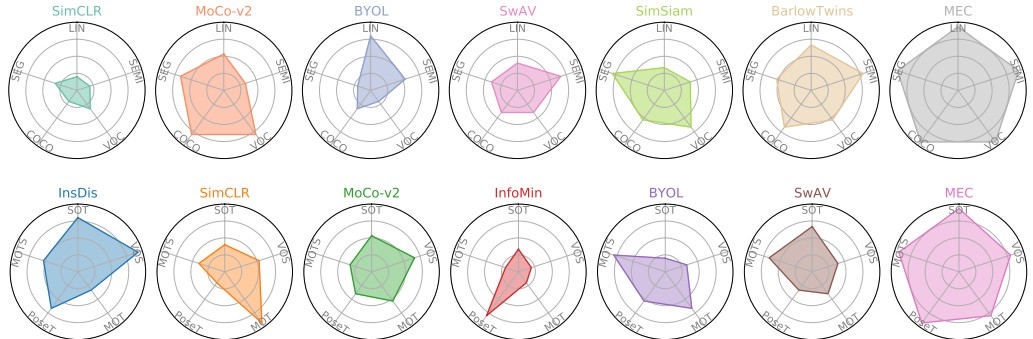

Figure 1: Comparison of transfer learning performance on five image-based tasks (top row) and five video-based tasks (bottom row). Along each axis we plot the performance ranking of the representation on a specific downstream task, so a polygon with a larger area means better generalization capacity on various downstream tasks, across the board.

model to predict the rotation angles [27]. In both cases, to fulfill the pretext task, the model must learn a meaningful representation that can describe object texture, shape or even category, therefore the learned representation transfers well to downstream tasks related to these features.

Despite successes of existing pretext tasks, we find that they inevitably introduce biases into the learned representation, which conflicts with the original aim of "general-purpose". For example, representations trained with the most prevalent image-level task, instance discrimination [20, 80], are found biased to image-level tasks such as image classification, while by contrast degenerate in patch- or pixel-level tasks like object detection and semantic segmentation [81]. Even being transferred to image-level tasks, such representations still suffer from domain gaps in cases of different data distributions [26], *e.g.*, classification on unseen categories other than ImageNet objects.

In this work, we are curious of *what makes for generalizable representations*, and pursue an explicit optimization with a criterion that directly measures the structure of representations, with the aim of minimizing the biases brought by the pretext task. To this end, we propose Maximum Entropy Coding (MEC). Inspired by the principle of maximum entropy in information theory, the basic hypothesis in MEC is that a generalizable representation should be the one that admit *the maximum entropy* among all plausible representations. Accordingly, optimizing towards maximum entropy leads to representations with good generalization capacity. The main challenge confronting us is that it is difficult, and computationally expensive, if possible, to estimate the distribution of a given representation (usually a finite set of high-dimensional vectors), so in turn it is difficult to estimate the entropy. To cope with this issue, we replace the optimization objective from the originally defined entropy to a computationally tractable surrogate, *i.e.*, the necessary number of bits needed to encode the representations via lossy data coding [16]. The log-determinant term costs the most computation in the coding length function. By leveraging Taylor expansion of matrix, we further approximate this term with polynomial functions, leading to significant speed up and thus makes large-scale pre-training possible.

In contrast to previous SSL methods that mainly evaluate on ImageNet classification, we experiment on a wide variety of vision tasks to show the good generalization capacity of MEC. The considered tasks span across not only image-level recognition on various data distributions, but also patch- or pixel-level tasks like object detection, instance segmentation, and object tracking. We show MEC generalizes well *consistently* across all tasks considered, while representations learned with previous pretext tasks usually perform well on closely related downstream tasks but degenerate on less related ones (see Figure 1). Besides empirical results, we find interesting equivalence between low-order approximations of MEC and existing batch-wise (*e.g.*, SimSiam [14]) or feature-wise objectives (*e.g.*, Barlow Twins [86]), which provides a new perspective for a unified understanding of prevalent SSL methods. In summary, the main contributions of our work are as follows:

- To learn representations generalizable to various downstream tasks, we introduce the principle of maximum entropy into self-supervised learning.

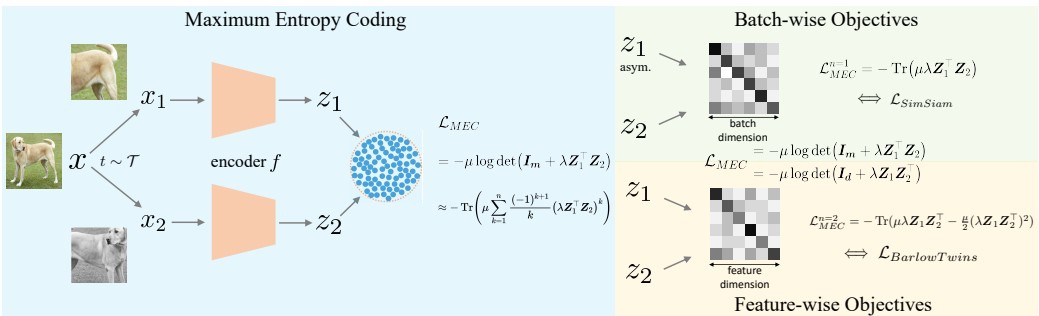

Figure 2: Illustration of our MEC and its relation to batch-wise and feature-wise objectives.

- We propose Maximum Entropy Coding (**MEC**), which explicitly optimizes on representations based on the principle of maximum entropy, and leverages the minimal coding length in lossy data coding as a computationally tractable surrogate for the entropy.
- We reformulate the log-determinant term in coding length function into a scalable form, which makes large-scale pre-training possible, and unifies existing batch-wise and feature-wise objectives as low-order approximations of our method.
- We show MEC representation generalizes well on a wide variety of image- and video-based downstream tasks, achieving state-of-the-arts on most tasks considered.

## 2 Method

In this section, we start with illustrating the maximum entropy principle, and then introduce a computationally tractable surrogate of the information entropy for high-dimensional vectors. We then present a scalable reformulation of the proposed surrogate, which makes large-scale training possible. And we further incorporate the view consistency prior for maximum entropy coding. Finally, we demonstrate how the proposed method can unify existing batch-wise and feature-wise SSL objectives. Please refer to Appendix E for more details about the proofs in this section.

### 2.1 Maximum Entropy Coding

**The maximum entropy principle.** The main purpose of this work is to improve the generalization capacity of self-supervised learning representations across unseen downstream tasks and data distributions, and reduce the biases brought by specifically designed pretext tasks as much as possible. This naturally raises a question, *i.e.*, what makes for a generalizable representation? To answer this question, we are particularly inspired by the maximum entropy principle in information theory, which states that the probability distribution that best represents the current state of knowledge about a system is the one with largest entropy, given a testable information (such as accuracy) and in this way no additional bias or assumptions is introduced [38, 39, 59]. We therefore hypothesis that a generalizable representation is the one that has the maximum entropy among all plausible representations. Intuitively, if we are able to express the entropy in a closed form, the maximum entropy principle then can serve as an optimization objective and supervise the representation learning.

**Minimal coding length as a surrogate for entropy.** Entropy is originally defined on probability distributions [63], *i.e.*, $H(z) \triangleq - \int p(z) \log p(z) dz$, for continuous random variables. However, it is very difficult to estimate the true distributions $p(z)$ of a representation [6, 54], from a finite set of high dimensional vectors $Z = [z^1, z^2, \ldots, z^m] \in \mathbb{R}^{d \times m}$. A handy fact is that entropy is conceptually equivalent to the minimal number of bits required to encode the data losslessly, so the minimal lossless coding length could be used to represent the entropy. However, lossless coding of continuous random variables is infeasible in our case since it often requires an infinite number of bits, breaking the numerical stability. Instead, we exploit the coding length in lossy data coding [16] as a computationally tractable surrogate for the entropy of continuous random variables. Given a set of samples $Z$, the minimal number of bits needed to encode $Z$ subject to a distortion $\epsilon$ is given by the

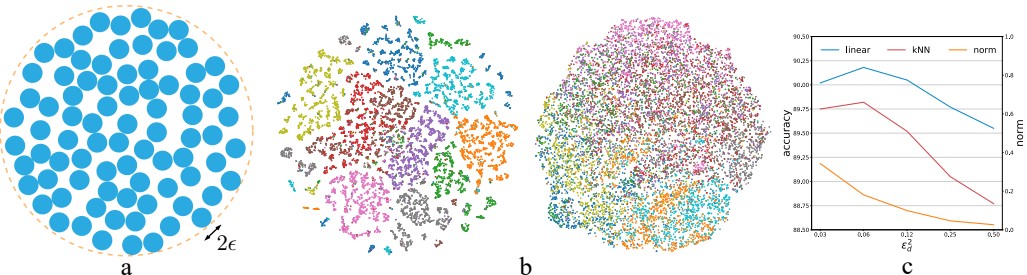

a                 b                 c

Figure 4: **Effects of the distortion measure $\epsilon$ on maximum entropy coding (MEC).** (a): Encoding the representations is akin to packing $\epsilon$-balls into the representation space. (b): T-SNE [71] visualization of the representations learned with large $\epsilon$ (left plot, $\epsilon_d^2 = 0.12$) and small $\epsilon$ (right plot, $\epsilon_d^2 = 0.01$). (c): Linear and kNN accuracy and the spectral norm *w.r.t* the degree of distortion $\epsilon$.

following coding length function [47, 72]:

$$L \triangleq \left( \frac{m+d}{2} \right) \log \det \left( \boldsymbol{I_m} + \frac{d}{m\epsilon^2} \boldsymbol{Z}^\top \boldsymbol{Z} \right), \tag{1}$$

where $\boldsymbol{I_m}$ denotes the identity matrix with dimension $m$, and $\epsilon$ is the upper bound of the expected decoding error between $z \in \boldsymbol{Z}$ and the decoded $\widehat{z}$, *i.e.*, $\mathbb{E}\left[\|z - \widehat{z}\|_2\right] \leq \epsilon$.

We note that the computation of log-determinant of high dimensional matrix in Equation (1) is highly expensive and may cause numerically unstable results for ill-conditioned matrix, which inhibits its application to large-scale pre-training (*e.g.*, over 1 million images). Therefore, a scalable and stable reformulation of Equation (1) is required. We first rewrite Equation (1) as $L = \mu \log \det \left( \boldsymbol{I_m} + \lambda \boldsymbol{Z}^\top \boldsymbol{Z} \right)$, where $\mu = \frac{m+d}{2}$ and $\lambda = \frac{d}{m\epsilon^2}$. Utilizing the identical equation $\det(\exp(\boldsymbol{A})) = \exp(\text{Tr}(\boldsymbol{A}))$ [35], we obtain $L = \text{Tr}\left(\mu \log \left( \boldsymbol{I_m} + \lambda \boldsymbol{Z}^\top \boldsymbol{Z} \right)\right)$, where $\text{Tr}$ stands for the trace of the matrix. Finally, we apply Taylor series expansion to expand the logarithm of the matrix and obtain

$$L = \text{Tr}\left( \mu \sum_{k=1}^{\infty} \frac{(-1)^{k+1}}{k} \left( \lambda \boldsymbol{Z}^\top \boldsymbol{Z} \right)^k \right), \tag{2}$$

with convergence condition: $\left\| \lambda \boldsymbol{Z}^\top \boldsymbol{Z} \right\|_2 < 1$, and this can be achieved by adjusting the hyperparameter $\epsilon$ (detailed in Section 2.2). Compared to Equation (1), there are only matrix multiplication and addition in Equation (2), which significantly speeds up the computation process and avoids the numerical unstability. To verify this, in Figure 3, we make a comparison of running time and relative approximation error between Equation (1) and (2) (using the first four terms). The results show that our reformulation can approximate the original coding length function with negligible error (all errors are well below $0.5\%$), and accelerate the computation considerably (over $50\times$ acceleration for all cases), thus making large-scale pre-training possible.

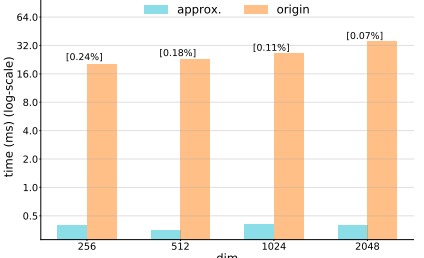

Figure 3: Comparison of running time and relative approximation error between Equation (1) (origin) and Equation (2) (approx.) for different number of samples in $\boldsymbol{Z}$ (dim).

## 2.2 Combining with the View Consistency Prior

It should be noted that the necessary premise of the maximum entropy principle is that *testable information* is given as prior. For example, the testable information could be the accuracy of a predictive model: the most generalizable model should be the one with maximum entropy, but it is only when a group of models reaches a given accuracy. Otherwise, simply optimizing towards maximum entropy will lead to trivial solutions such as uniform distributions. In Equation (1), the prior is not considered. To introduce a prior, we employ a common practice in SSL by augmenting $\boldsymbol{Z}$ into two different views $\boldsymbol{Z_1}$ and $\boldsymbol{Z_2}$. More specifically, given a

set of images $\mathcal{D}$, an image $x$ is sampled uniformly from $\mathcal{D}$, and two augmented views $x_1$ and $x_2$ are obtained from $x$ via a distribution of data augmentations $\mathcal{T}$. Then they are fed to an encoder $f$, consisting of a backbone and a projector network, which produces $\ell_2$-normalized embeddings of $z_1$ and $z_2$. For a batch of $m$ images, we have $\boldsymbol{Z_1} = \left[ z_1^1, z_1^2, \ldots, z_1^m \right]$ and similarly for $\boldsymbol{Z_2}$, which are two observations of the same $\boldsymbol{Z}$. MEC aims to minimize the following loss:

$$\mathcal{L}_{MEC} = -\mu \log \det \left( \boldsymbol{I_m} + \lambda \boldsymbol{Z}_1^\top \boldsymbol{Z}_2 \right) \approx -\operatorname{Tr} \left( \mu \sum_{k=1}^{n} \frac{(-1)^{k+1}}{k} \left( \lambda \boldsymbol{Z}_1^\top \boldsymbol{Z}_2 \right)^k \right), \qquad (3)$$

where the same notations in Section 2.1 apply, and $n$ is the order of Taylor expansion. Compared with Equation (1), the formulation in (3) considers not only maximizing entropy, but also the view consistency prior mined from the data itself, therefore learning meaningful representations.

As noted in Section 2.1, the convergence condition of Taylor expansion requires $\|\boldsymbol{C}\|_2 < 1$, where $\boldsymbol{C} = \lambda \boldsymbol{Z}_1^\top \boldsymbol{Z}_2$ and $\lambda = \frac{d}{m\epsilon^2} = \frac{1}{m\epsilon_d^2}$. We show such condition can be strictly satisfied by setting $\epsilon_d^2 > 1$ because of the inequality $\|\boldsymbol{C}\|_2 \leq \sqrt{\|\boldsymbol{C}\|_1 \|\boldsymbol{C}\|_\infty} < 1$. In practice, we empirically find that the Taylor expansion converges over a wide range of $\epsilon_d$ (Figure 4(c)) with a linear warm-up. From the preliminary experiments on CIFAR-10 [41] (detailed in Appendix B), we also find that the distributions of representations show progressive finer granularity as $\epsilon_d$ decreases (Figure 4(b)). This can be interpreted by the practical meaning of the distortion $\epsilon_d$ (Figure 4(a)), *i.e.*, a smaller $\epsilon_d$ encourages the representation space to be encoded in finer granularity (and hence more uniform). By contrast, a small $\epsilon_d$ might break the semantic structures of similar images (*i.e.*, tolerance). Therefore, a good choice of $\epsilon_d$ is needed to compromise the uniformity and tolerance properties [75, 74] of representations, which shares the same role as the temperature [80] term in contrastive learning.

An overview of MEC is illustrated in Figure 2 and a PyTorch-like pseudocode is provided in Appendix A. The algorithm describes the minimalist variant of MEC, which can be further improved by integrating momentum encoder and asymmetric networks (detailed in experiments).

## 2.3 A Unified View of Batch-wise and Feature-wise SSL Objectives

Current SSL methods based on Siamese networks can be roughly divided into two categories: batch-wise methods [12, 31, 13, 15, 14, 10] and feature-wise methods [86, 5, 24, 36]. The former aims to minimize the distance between augmented views of the same sample while maximizing the distance between different samples, which can be viewed as decorrelating the different features in a batch. The latter, in contrast, tries to decorrelate the different vector components in the representation. The relationship between them has not been fully understood. Our work builds bridges between these two types of methods through the following derivation:

$$\mathcal{L}_{MEC} = \underbrace{-\mu \log \det \left( \boldsymbol{I_m} + \lambda \boldsymbol{Z}_1^\top \boldsymbol{Z}_2 \right)}_{\text{batch-wise}} = \underbrace{-\mu \log \det \left( \boldsymbol{I_d} + \lambda \boldsymbol{Z}_1 \boldsymbol{Z}_2^\top \right)}_{\text{feature-wise}}, \qquad (4)$$

which can be proved since $\boldsymbol{Z}_1^\top \boldsymbol{Z}_2 \in \mathbb{R}^{m \times m}$ and $\boldsymbol{Z}_1 \boldsymbol{Z}_2^\top \in \mathbb{R}^{d \times d}$ have the same nonzero eigenvalues. In Figure 2, under the framework of MEC, we show the equivalence between batch-wise and feature-wise methods using two examples, SimSiam [14] and Barlow Twins [86]. By taking Taylor expansion (Equation (2)) of the left side of Equation (4) and before the trace operation, the diagonal elements of the leading term (*i.e.*, $\mu\lambda \boldsymbol{Z}_1^\top \boldsymbol{Z}_2$) measure the similarity between the views of the same images in a batch, and the objective of SimSiam [14] is equivalent to maximizing the trace of this term. Similarly, the leading term of the right side expansion models the correlation between dimensions of the feature, and the objective of Barlow Twins [86] is equivalent to the second-order expansion of $\mathcal{L}_{MEC}$. With the above derivation, our method naturally subsumes the two different kinds of objectives as its low-order expansions, and we show in experiments that better downstream task performance can be achieved with higher-order approximations. We further show in Appendix E that our MEC can also bridge other self-supervised objectives. And we hope the direct tying of a family of objectives to a very grounded mathematical concept can inspire more new methods.

## 3 Experiments

We perform self-supervised pre-training using the proposed MEC on the training set of the ImageNet ILSVRC-2012 dataset [17]. After pre-training, we conduct extensive experiments to examine

Table 1: **Linear evaluation**. All methods are based on standard ResNet-50 [33] pre-trained with two 224×224 views on ImageNet training dataset. We perform pre-training with four different length of epochs following [14].

| Method | Pre-training epochs | | | |
|---|---|---|---|---|
| | 100 | 200 | 400 | 800 |
| SimCLR [12] | 66.5 | 68.3 | 69.8 | 70.4 |
| MoCo v2 [13] | 67.4 | 69.9 | 71.0 | 72.2 |
| BYOL [29] | 66.5 | 70.6 | 73.2 | 74.3 |
| SwAV [10] | 66.5 | 69.1 | 70.7 | 71.8 |
| SimSiam [14] | 68.1 | 70.0 | 70.8 | 71.3 |
| Barlow [86] | 67.3 | 70.2 | 71.8 | 73.0 |
| **MEC** | **70.6** | **71.9** | **73.5** | **74.5** |

Table 2: **Semi-supervised classification**. We finetune the pre-trained model using 1% and 10% training samples of ImageNet following [12, 29], and the top-1 and top-5 accuracy on ImageNet val dataset are reported.

| Method | 1% | | 10% | |
|---|---|---|---|---|
| | Top 1 | Top 5 | Top 1 | Top 5 |
| Supervised | 25.4 | 48.4 | 56.4 | 80.4 |
| SimCLR [12] | 48.3 | 75.5 | 65.6 | 87.8 |
| BYOL [29] | 53.2 | 78.4 | 68.8 | 89.0 |
| Barlow [86] | 55.0 | 79.2 | 69.7 | 89.3 |
| DINO [11] | 52.2 | 78.2 | 68.2 | 89.1 |
| VICReg [5] | 54.8 | 79.4 | 69.5 | 89.5 |
| **MEC** | **55.9** | **79.6** | **70.3** | **89.7** |

Table 3: **Transfer learning on object detection and instance segmentation tasks**. We fine-tune the pre-trained model end-to-end on target datasets and tasks, following the standard protocol described in [13, 14, 29, 86]. We use Faster R-CNN [61] for VOC detection tasks and Mask R-CNN [32] (1× schedule) for COCO detection and instance segmentation tasks. All Faster/Mask R-CNN models are with the C4-backbone [79]. **Bold entries** are within 0.2 below the best.

| Pre-train | VOC 07 detection | | | VOC 07+12 detection | | | COCO detection | | | COCO instance seg. | | |
|---|---|---|---|---|---|---|---|---|---|---|---|---|
| | $AP_{50}$ | AP | $AP_{75}$ | $AP_{50}$ | AP | $AP_{75}$ | $AP_{50}$ | AP | $AP_{75}$ | $AP_{50}^{mask}$ | $AP^{mask}$ | $AP_{75}^{mask}$ |
| Scratch | 35.9 | 16.8 | 13.0 | 60.2 | 33.8 | 33.1 | 44.0 | 26.4 | 27.8 | 46.9 | 29.3 | 30.8 |
| Supervised | 74.4 | 42.4 | 42.7 | 81.3 | 53.5 | 58.8 | 58.2 | 38.2 | 41.2 | 54.7 | 33.3 | 35.2 |
| SimCLR [12] | 75.9 | 46.8 | 50.1 | 81.8 | 55.5 | 61.4 | 57.7 | 37.9 | 40.9 | 54.6 | 33.3 | 35.3 |
| MoCo v2 [13] | 77.1 | **48.5** | **52.5** | 82.5 | **57.4** | 64.0 | 58.9 | 39.3 | 42.5 | 55.8 | 34.4 | 36.5 |
| BYOL [29] | 77.1 | 47.0 | 49.9 | 81.4 | 55.3 | 61.1 | 57.8 | 37.9 | 40.9 | 54.3 | 33.2 | 35.0 |
| SwAV [10] | 75.5 | 46.5 | 49.6 | **82.6** | 56.1 | 62.7 | 58.6 | 38.4 | 41.3 | 55.2 | 33.8 | 35.9 |
| Barlow [86] | 75.7 | 47.2 | 50.3 | **82.6** | 56.8 | 63.4 | 59.0 | 39.2 | 42.5 | 56.0 | 34.3 | 36.5 |
| SimSiam [14] | **77.3** | **48.5** | **52.5** | 82.4 | 57.0 | 63.7 | 59.3 | 39.2 | 42.1 | 56.0 | 34.4 | **36.7** |
| **MEC** | **77.4** | 48.3 | 52.3 | **82.8** | **57.5** | **64.5** | **59.8** | **39.8** | **43.2** | **56.3** | **34.7** | **36.8** |

the learned representations on various tasks. The tasks considered include four image-level tasks: ImageNet linear probing, ImageNet semi-supervised classification, object detection, and instance segmentation; and five video-based tasks: single object tracking (SOT) [78], video object segmentation (VOS) [56], multi-object detection (MOT) [48], multi-object detection and segmentation (MOTS) [73] and pose tracking (PoseTrack) [1]. An overview comparison with existing SSL representations across different tasks is presented in Section 3.1. Moreover, in Section 3.4, we present extensive ablations studies to demonstrate the behaviors and properties of the proposed method. All the implementation details of the aforementioned experiments are deferred to Appendix C.

## 3.1 Overview Comparison

We evaluate MEC together with multiple previous SSL representations on the four image-based tasks and the five video-based tasks, plot their performance ranking along a axis for a specific task and obtain two sets of radar-like charts in Figure 1. Intuitively, it can be seen that the larger a polygon in a radar chart is, the better the corresponding representation generalize across downstream tasks. MEC consistently outperforms previous representations on both image- and video-based tasks considered, suggesting its good generalization capacity to various downstream tasks and data distributions. Below we detail experiments on each task. And additional experiment results can be found in Appendix D, including pre-training with different strategies and datasets, transfer learning on fine-grained classification benchmarks.

## 3.2 Evaluation on ImageNet

**Linear probing.** We train a linear classifier on top of frozen representations of the pre-trained model on the ImageNet training set, and report the top-1 accuracy on the ImageNet validation set, which is a

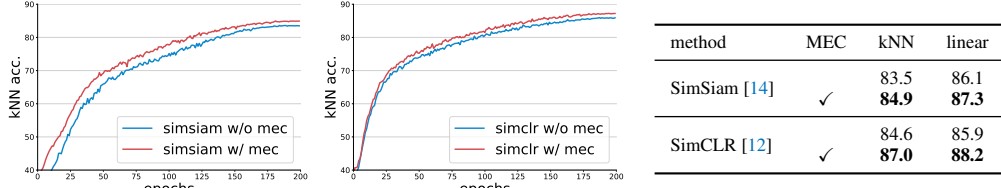

Figure 5: **Effects of MEC regulation on other SSL objectives on CIFAR-10 [41]**. Left plot: accuracy of a kNN classifier as a monitor of the pre-training process. Right table: linear probing and kNN classification accuracy of the pre-trained models with *v.s.* without our MEC regulation.

standard and important protocol in SSL [88, 12, 31, 14, 86]. We evaluate the learned representations of models with four different pre-training epochs following [14], and make comparisons in Table 1 with state-of-the-art methods, whose results are reported in [14] and better than their original papers. Even so, our method MEC achieves highest accuracy under all pre-training epochs among those methods. In particular, with only 200-epoch pre-training, MEC outperforms several 800-epoch pre-trained methods, including SimCLR [12], SwAV [10] and SimSiam [14], which indicates the pre-training efficiency of our method. Additionally, by incorporating other pre-training strategies, such as multi-crop [10, 11], our method can attain even better results, which are detailed in Appendix D. Furthermore, for a fixed given downstream task (*i.e.*, image classification), MEC also improves generalization across different data distributions, which can be concluded from the transfer learning experiments on 11 fine-grained classification datasets in Appendix D.

**Semi-supervised classification.** We fine-tune the pre-trained model on a small subset of ImageNet for classification task. Specifically, we adopt the same fixed splits of 1% and 10% of ImageNet training set as in [12] and report both top-1 and top-5 accuracies in Table 2. The experiment results show that our method consistently outperforms previous approaches in both 1% and 10% settings. And we also note that all SSL methods surpass the supervised counterparts by a large margin (*e.g.*, over 20% accuracy using 1% labeled data), which emphasizes the importance of leveraging large amount of unlabeled data.

### 3.3  Transfer learning

**Object detection and instance segmentation.** We transfer the pre-trained model by end-to-end fine-tuning on object detection and instance segmentation tasks, following the common practice of previous methods [31, 13, 14, 29, 86]. We report the results in Table 3. Similar to the observations in semi-supervised classification, the SSL methods outperform or are on par with the supervised counterparts in all tasks. Our method achieves better performance than previous approaches in most of the tasks and metrics considered. In particular, MEC is significantly better than the supervised baseline (+1.6 AP), SimCLR (+1.9 AP), and previous best method MoCo v2 (+0.5 AP) on COCO detection task. These results validate that the model pre-trained by MEC can generalize very well across different image-based tasks and datasets.

**Video-based tasks.** To further evaluate whether MEC's representation generalizes beyond image-based tasks, we transfer the pre-trained model to a series of video tasks based on a recently proposed evaluation platform named UniTrack [76]. In contrast to above evaluation processes, UniTrack does not require any additional training and hence provides a more direct comparison between different representations. We report the results in Table 4 and compare different methods using radar plots in Figure 1. As noted in UniTrack, there is no significant correlation between linear probe accuracy and tracking performance. However, our method achieves strong performance across linear probing and all five video tracking tasks. Specifically, MEC achieves the highest rank on four our of five tasks, and ranks among the top 2 methods on seven metrics out of all nine metrics. These results substantiate that the learned representations of MEC are more general-purpose and versatile. We attribute this important property of MEC to the principle of maximum entropy, the basis of our method, which enables the model to make best use of data for unseen downstream tasks during pre-training.

Table 4: **Transfer learning on five video tracking tasks.** All methods use the same ResNet-50 backbone [33] and are evaluated based on UniTrack [76]. For each cell, we report the results obtained from features at [layer3 / layer4], and the best performance between the two is **bolded**. We then use the best of the two to rank the top 2 models (underlined) in each column.

| Pre-train | SOT [78] | | VOS [56] | MOT [48] | | MOTS [73] | | PoseTrack [1] | |
|---|---|---|---|---|---|---|---|---|---|
| | $AUC_{XCorr}$ ↑ | $AUC_{DCF}$ ↑ | $\mathcal{J}$-mean↑ | IDF1↑ | HOTA↑ | IDF1↑ | HOTA↑ | IDF1↑ | IDs↓ |
| Rand. Init. | **10.3** / 9.0 | **28.0** / 20.0 | 29.3 / **33.9** | 8.4 / **8.9** | 8.4 / **8.5** | 20.8 / **23.1** | 25.9 / **28.7** | **40.2** / 38.5 | **88792** / 90963 |
| Supervised | **58.6** / 49.5 | **62.0** / 53.9 | **62.3** / 57.9 | **75.6** / 73.2 | **63.3** / 61.8 | 68.4 / **69.4** | 70.2 / **71.0** | **73.7** / 73.3 | **6969** / 7103 |
| InsDis [80] | 47.6 / 47.3 | **61.8** / 51.1 | **62.6** / 60.1 | 66.7 / **73.9** | 57.9 / **61.9** | **68.4** / 68.0 | 69.6 / **70.3** | 72.4 / **73.9** | 7106 / **7015** |
| MoCo v1 [31] | 50.9 / 47.9 | **62.2** / 53.7 | **61.5** / 57.9 | 69.2 / **74.1** | 59.4 / **61.9** | **70.6** / 69.3 | **71.6** / 70.9 | 72.8 / **73.9** | **6872** / 7092 |
| SimCLR [12] | 47.3 / **51.9** | **61.3** / 50.7 | 60.5 / 56.5 | 66.9 / **75.6** | 57.7 / **63.2** | 65.8 / **67.6** | 67.7 / **69.5** | 72.3 / **73.5** | **7084** / 7367 |
| MoCo v2 [13] | **53.7** / 47.2 | **61.5** / 53.3 | **61.2** / 54.0 | 72.0 / **74.9** | 61.2 / **62.8** | **67.5** / 67.3 | 69.6 / **69.6** | 73.0 / **73.7** | **6932** / 7702 |
| Infomin [68] | **48.5** / 46.8 | **61.2** / 51.9 | **58.4** / 51.1 | 66.7 / **73.4** | 57.6 / **61.9** | **66.7** / 66.3 | 68.5 / **68.8** | 72.5 / **74.0** | **7066** / 7901 |
| BarLow [86] | 44.5 / **55.5** | **60.5** / 60.1 | **61.7** / 57.8 | 63.7 / **74.5** | 55.4 / **62.4** | **68.7** / 67.4 | 69.5 / **69.8** | 72.3 / **74.3** | **7131** / 7456 |
| BYOL [29] | 48.3 / **55.5** | **58.9** / 56.8 | **58.8** / 54.3 | 65.3 / **74.9** | 56.8 / **62.9** | **70.1** / 66.8 | **70.8** / 69.3 | 72.4 / **73.8** | **7213** / 8032 |
| SwAV [10] | 49.2 / **52.4** | **61.5** / 59.4 | **59.4** / 57.0 | 65.6 / **74.4** | 56.9 / **62.3** | **68.8** / 67.0 | **69.9** / 69.5 | 72.7 / **73.6** | **7025** / 7377 |
| PixPro [82] | 40.5 / **49.2** | **57.4** / 49.3 | **56.4** / 52.2 | 61.7 / **67.7** | 54.3 / **58.6** | 64.2 / **66.2** | 65.1 / **67.6** | 72.4 / **73.1** | 7163 / **6953** |
| DetCo [81] | **55.0** / 47.1 | **59.0** / 53.2 | **62.3** / 56.1 | **75.3** / 72.9 | **62.8** / 61.6 | **67.8** / 66.8 | **70.0** / 69.4 | **73.9** / 73.3 | **7357** / 8009 |
| **MEC** | **56.8** / 54.7 | **62.3** / 60.6 | **62.0** / 57.5 | 72.7 / **75.0** | 61.2 / **63.3** | **70.7** / 66.5 | **72.0** / 69.0 | 73.1 / **74.1** | **6665** / 7752 |

Table 5: **MEC ablation experiments with ResNet-50 on ImageNet.** We report the results of linear probing. The default settings of these experiments are marked in $\boxed{\text{gray}}$.

(a) Smaller epochs.

| epochs | 30 | 50 | 70 | 100 |
|---|---|---|---|---|
| acc. | 62.1 | 67.5 | 69.2 | 70.6 |

(b) Smaller batch sizes.

| batch | 128 | 256 | 512 | 1024 |
|---|---|---|---|---|
| acc. | 69.5 | 70.1 | 70.4 | 70.6 |

(c) The distortion $\epsilon_d$.

| $\epsilon_d^2$ | 0.02 | 0.06 | 0.25 | 1.0 |
|---|---|---|---|---|
| acc. | 70.5 | 70.6 | 70.2 | 69.5 |

(d) The momentum coefficient.

| m | 0 | 0.9 | 0.99 | 0.996 | 0.999 |
|---|---|---|---|---|---|
| acc. | 69.0 | 70.1 | 70.4 | 70.6 | 70.2 |

(e) The order of expansion.

| order | 1 | 2 | 3 | 4 | 5 |
|---|---|---|---|---|---|
| acc. | 70.0 | 70.3 | 70.5 | 70.6 | 70.6 |

(f) The projector network.

| proj | asym. | sym. | 4096 | 8192 |
|---|---|---|---|---|
| acc. | 70.6 | 70.1 | 70.7 | 71.2 |

## 3.4 Ablation studies

**Batch size and training epoch.** The computation and memory resources required for training self-supervised models are demanding. Previous methods often need a large batch size (*e.g.*, 4096 in SimCLR [12]) and long training time (*e.g.*, 1000 epochs in Barlow Twins [86]) to work well. We hence test the efficiency of our method by pre-training the model with smaller training epochs (Table 5a) and batch sizes (Table 5b). The results show that our method can effectively avoid those prohibitive requirements. In particular, Our 50-epoch pre-trained model has already outperformed those 100-epoch pre-trained models of other methods (*e.g.*, SimCLR [12], BYOL [29], Barlow Twins [86]). Moreover, MEC is more robust to different batch sizes, and even a batch size of 128 performs well (with 1.1% accuracy drop). These results are noticeably different from those methods based on the pretext task of instance discrimination, where a large number of negative samples is required (*e.g.*, over 4% accuracy drop of SimCLR with a batch size of 128). This behavior also liberates MEC from those complex designs, such as memory queue [13] and online clustering [10].

**Siamese networks** play a critical role in discriminative SSL, and different methods often rely on different architectures to prevent mode collapse. We investigate the effects of different designs of Siamese networks by varying the momentum coefficient (Table 5d) and the projector network (Table 5f). Compared to other methods [13, 14, 29, 86], MEC still works well when the Siamese networks are direct weight-sharing (m=0) and symmetric (sym.). In contrast, BYOL [29] requires the momentum encoder to prevent collapse (0.3% accuracy without it), and asymmetric architectures are essential for SimSiam [14] to work. These results demonstrate that the objective of MEC naturally avoids representational collapse without relying on special design of Siamese networks. We also observe that the performance can be further improved with a larger projector network, consistent with the conclusions in previous methods [86, 5].

**Taylor series approximation.** The computation of log-determinant of a high-dimensional matrix can be significantly accelerated by exploiting Taylor series approximation. And we study two important hyper-parameters of the process, *i.e.*, the order of Taylor expansion (Table 5e) and the distortion

Table 6: **Comparison of SSL methods with different architectures.** (ConvNets *v.s.* Transformer) All methods are pre-trained with two 224×224 crops on ImageNet training set, and we report the results of linear probing on ImageNet validation set.

| model | **MEC** | iBOT [90] | MAE [30] | DINO [11] | MoCo v3 [15] | SimCLR [12] | BYOL [29] | SwAV [10] |
|---|---|---|---|---|---|---|---|---|
| R-50, 800-ep | **74.5** | - | - | - | 73.8 | 70.4 | 74.3 | 71.8 |
| ViT-S, 300-ep | **73.4** | 73.2 | 68.2 | 70.9 | 72.5 | 69.0 | 71.0 | 67.1 |
| ViT-B, 300-ep | **76.5** | 75.6 | 69.3 | 72.6 | **76.5** | 73.9 | 73.9 | 71.6 |

measure $\epsilon_d$ (Table 5c). We find that the accuracy increases with higher order of expansion, since it produces more accurate approximation. Besides the linear probing, we find the COCO detection performance can be improved by a large margin (+1.5 AP) when the order is increased from 1 to 4. We also note that the extra two orders (from 2 to 4) benefit less than lifting from the first order to the fourth order approximation. It is reasonable since the relative approximation error of second-order expansion is already quite decent, lower than 0.5% (Figure 6), almost the same as fourth-order expansion (Figure 3). As for the distortion hyper-parameter, on one hand, it should be large enough so that the Taylor expansion is convergent, and we empirically observe this condition is satisfied over a wide range of degree of distortion. On the other hand, a good choice of $\epsilon_d$ is also needed to improve the representation quality, because it compromises the uniformity and tolerance properties of representations as depicted in Figure 4 and discussed in Section 2.2.

**Generalization across different backbones and objectives.** Given the recent progress in Vision Transformers (ViTs) [21], it is worthwhile to examine the generalization ability of our method on both ViTs and ConvNets, though previous methods usually focus on only one architecture. To this end, we make a straightforward extension by replacing the ConvNet encoder with a ViT encoder, following the practice of [15]. The comparisons are made in Table 5 with previous methods based on Siamese networks [12, 29, 10, 11, 31], and also recently proposed masked auto-encoding methods [30, 90] tailored for Transformers. And the results show that MEC performs equally well with both architectures. Furthermore, since MEC is designed for optimizing the representation quality, it should be complementary to other methods that focus on pretext tasks. To validate this hypothesis, we apply MEC as a regulation term on the representations of a contrastive method, SimCLR [12], and a non-contrastive method, SimSiam [14]. The results in Figure 5 demonstrate that MEC improves both linear and kNN accuracy by a large margin (*e.g.*, 2.3% linear probing accuracy for SimCLR) with negligible computation overhead (~1.05×running time).

## 4 Related Work

**Maximum entropy principle** states that the probability distribution which best represents the current state of knowledge is the one with largest entropy, in the context of precisely stated prior data. This principle originates in Laplace's "Principle of Insufficient Reason" and is popularized by Jaynes [38, 39]. It has a wide variety of applications in different fields [40], ranging from statistical physics [58, 51], economics [2, 62], to reinforcement learning [77, 49] and supervised learning [57, 64, 22]. To the best of our knowledge, the usage of this principle has not been explored in the context of SSL. We notice a closely related work MCR$^2$ [85] exploits the same coding length function (Equation (1)), but the motivation behind is quite different: they do not originate from maximum entropy, but instead base their method on coding rate reduction. Furthermore, MCR$^2$ experiments on classification and clustering tasks, and is difficult to generalize to modern SSL settings due to the expensive and unstable computation. In contrast, our method is tailored for SSL and we reformulate the coding length function into a scalable and stable form for large-scale pre-training. Another closely related concept in information theory is mutual information, which has been widely used in SSL [34, 53, 68, 67, 69] based on the infomax principle [45]. Our method can also be viewed as an application of the infomax principle that maximizes the representation information, but without using mutual information estimators, which might result in worse transfer performance [70].

**Pretext tasks** are one of the core components in SSL. They refer to a type of tasks that mine a certain structure of the data, and then force learners to predict such a structure from degenerated input data. Labels are mined from the data itself, therefore dispensing with the need of human annotations. In natural language processing, the most commonly used pretext task is masked language modeling (MLM) [18, 60], where a random portion of the input sequence is masked and the model learns to

predict these masked content. In computer vision, however, there emerge a much wider variety of pretext tasks [87, 88, 52, 83, 20, 80, 55, 19, 30, 4, 42], since images contain much more information than languages so that there are much more intrinsic structures to be mined as free learning signals Solving pretext tasks requires the model to understand the visual concepts present in images and thus useful representations can be learned. In this work, however, we argue that those handcrafted pretext tasks, like handcrafted feature engineering methods, might introduce undesired bias into the representations. So we pursue a straightforward approach to optimize on representations. And considering that the ultimate goal of self-supervised learning is a general-purpose representation, we explicitly encourage the generalization ability on downstream tasks and minimize the bias in the formulation of the pretext task, by introducing the maximum entropy principle.

**Siamese representation learning** uses two identical neural networks to learn representations by comparing entities, in supervised manner [8, 65, 7], or recently in self-supervised manner [12, 31, 13, 14, 29, 10, 11, 86, 5, 23, 43]. In Siamese SSL, the two branches are weight-sharing [12, 14], or one is a moving average of the other [31, 29]. One crucial problem that needs to be carefully addressed in Siamese SSL is mode collapse, where all data is mapped to the same representation. Contrastive methods, such as SimCLR [12] and MoCo [31], prevent the trivial solution by minimizing the distance between augmented views of the same images while maximizing the distance between different images. Clustering-based methods [9, 3, 10] enforces consistency between cluster assignments of the views from same images, where the cluster centers work as negative prototypes. Non-contrastive methods [14, 29] remove the use of negative pairs and SimSiam [14] concludes that the stop-gradient operation is critical to preventing mode collapse. Our work also leverages the basic view consistency prior inherent in Siamese representation learning, which gives testable information for the application of maximum entropy principle. Compared to previous work, our method of maximum entropy coding naturally avoids the low-entropy mode collapse problem and can be easily implemented without requiring instance discrimination [20, 12], memory queue [80, 31, 13] or online clustering [9, 3, 10]. Furthermore, we find interesting equivalence between low-order approximations of MEC and existing batch-wise or feature-wise objectives, which provides a new perspective for a unified understanding of SSL methods.

## 5    Conclusion

*What makes for generalizable representations?* In this work, we argue that the best generalizable representation is supposed to be the one that admits the maximum entropy. Based on this assumption, we propose MEC, a novel pretext task that aims explicitly to generalize well. MEC pursues a representation with maximum entropy in all eligible ones that satisfies the view consistency prior of Siamese representation learning. The minimal coding length in lossy data coding is leveraged as a computationally tractable surrogate for the entropy; and in order to facilitate large-scale pre-training, we reformulate the coding length function into a scalable form. With the reformulation, we further demonstrate that MEC bridges the batch-wise and feature-wise SSL objectives. Extensive experiments shows MEC generalizes well across various image/video tasks, various data distributions, and different network architectures. Furthermore, MEC can be readily plugged into existing SSL methods such as SimCLR and SimSiam, and brings consistent improvements upon these strong representations. We hope our exploration will inspire the community to rethink the possibility of optimizing generalization explicitly in self-supervised representation learning.

## 6    Funding Transparency Statement

Acknowledgement: This work is supported by the state key development program in 14th Five-Year under Grant No. 2021YFF0602103, 2021YFF0602102, 2021QY1702. We also thank for the research fund under Grant No. 2019GQG0001 from the Institute for Guo Qiang, Tsinghua University.

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
