# Appendices: Self-Supervised Learning via Maximum Entropy Coding

## A  Pseudocode of MEC

---
**Algorithm 1** PyTorch-like pseudocode of MEC

---

```
# f: encoder consisting of a backbone and a projector
# mu: a constant related to m and d
# lamda: a hyperparameter determined by the distortion
# n: the order of Taylor expansion

for x in loader: # load a minibatch x with m samples
    x1, x2 = aug(x), aug(x) # augmentation
    z1, z2 = f(x1), f(x2) # l2 normalized embeddings: [m, d] each

    loss = mec(z1, z2, mu, lamda, n)
    loss.backward()
    update(f) # optimizer update of f

# the loss of mec
def mec(z1, z2, mu, lamda, n):
    c = lamda*mm(z1, z2.t()) # [m, m] batch-wise
    # c = lamda*mm(z1.t(), z2) # [d, d] feature-wise
    power = c
    sum_p = zeros_like(power)
    for k in range(1, n+1): # n>1 for symmetric nets
        if k > 1 :
            power = mm(power, c)
        if (k + 1) % 2 == 0:
            sum_p += power / k
        else:
            sum_p -= power / k
    loss = -mu * trace(sum_p)
    return loss
```

---

**Notes**: `mm` is matrix multiplication. `t()` is transpose.

## B  CIFAR-10 Experiments

In this section, we detail the setting of the preliminary experiment described in Figure 4 in the main text. We train two models with different hyperparameters $\epsilon = 0.12$ and $\epsilon = 0.01$. After training, we extract representations of the CIFAR-10 training set and employ T-SNE [71] to map the representation to a two-dimensional space for visualization. Besides the hyperparameter $\epsilon$, other training configurations are kept identical and detailed below. Following the practice in [14], we do not use blur augmentation, and adopt the CIFAR variant of ResNet-18 [33] as backbone, Specifically, we remove the first max-pooling layer of ResNet-18, and set the kernel size of the first convolution layer to 3. The last classification layer is also removed and we treat the features after global average pooling as inputs to the projector, which is a two-layer MLP with BN [37] and ReLU [50] applied. The dimension of the output representation is 2048. We use SGD optimizer with weight decay $=0.0005$, momentum $=0.9$, and set the base learning rate to 0.03, which is linearly scaled with the batch size of 256. The learning rate is scheduled to a cosine decay rate for 600 epochs.

## C  Implementation Details.

**Data augmentations.** We adopt the same set of data augmentations following the common practice of previous methods [12, 29, 86, 14, 15], which is composed of geometric, color, and blurring augmentations. The geometric augmentations include random cropping, resizing to $224 \times 224$, and random horizontal flipping. The color augmentations consist of a random sequence of brightness,

contrast, saturation, hue adjustments, and a grayscale conversion. The blurring augmentations include Gaussian blurring and solarization. We use the same augmentation parameters as BYOL [29] and Barlow Twins [86]. For each iteration, each image is augmented twice to generate two views according to the above augmentation policy.

**Architecture.** We use a standard ResNet-50 network [33] without the final classification layer as the backbone, which yields a feature with dimension of 2048. It is followed by a projector network, which is a three-layer MLP with BN [37] and ReLU [50] applied, and each with 2048 output units. A momentum encoder is utilized to stabilize the training and further improve the performance. We turn one branch of the Siamese networks as online network and the other as target network, whose parameters are an exponential moving average of the online parameters. For the asymmetric network design, we append a two-layer MLP only to the online branch, and it has hidden dimension 512 and output dimension 2048 with BN [37] and ReLU [50] applied to the first layer. The output embeddings of the two branches are fed to the objective function for self-supervised pre-training. And after pre-training, we only keep the encoder for downstream tasks.

**Optimization.** We use the SGD optimizer with a cosine decay learning rate schedule [46] and a linear warm-up period of 10 epochs. The weight decay is $1.0 \times 10^{-5}$ and the momentum is 0.9. We set the base learning rate to 0.5, which is scaled linearly [28] with a batch size of 256 (*i.e.*, LearningRate $= 0.5 \times$ BatchSize/256). The exponential moving average parameter is increased from 0.996 to 1 with a cosine scheduler. We set the level of distortion $\epsilon_d^2 = 0.06$ and use a batch size of 1024. To enable large-batch and faster pre-training of 800 epochs, we adopt the LARS optimizer [84] with a batch size of 4096 and set the base learning rate to 0.3 and weight decay to $1.5 \times 10^{-6}$. We use the 800-epoch pre-trained model for downsteam tasks. To give an intuition of the computation overhead of our method, it takes MEC 42 hours for 100-epoch pre-training on 8 V100 GPUs, while it takes BYOL and Barlow Twins 45 and 48 hours on the same hardware.

**Linear probing.** We adopt the standard linear probing protocol [12, 14, 29, 86] and train a supervised linear classifier on top of the frozen representation. We use the LARS optimizer [84] with a batch size of 4096 and a cosine learning rate schedule over 100 epochs. The momentum is 0.9 and the wight decay is set to 0. During training, the input images are augmented by taking a random crop, resizing to $224 \times 224$, and flipping horizontally. At test time, we resize the image to $256 \times 256$ and then center-crop it to a size of $224 \times 224$.

**Semi-supervised classification.** We follow the semi-supervised learning protocol of [12, 86, 29] and fine-tune the pre-trained model on the 1% and 10 % subset of ImageNet [17] training set, using the same splits as in SimCLR [12]. We use the SGD optimizer with a batch size of 1024 and a momentum of 0.9. And we set the weight decay to 0. We use a base learning rate of 0.05 and fine-tune the model for 50 epochs. The data augmentations are the same as in linear probing.

**Object detection and instance segmentation.** We follow the common practice of previous methods [31, 13, 14, 86] and evaluate the transfer learning performance based on Detectron2 library [79]. We initialize the backbone ResNet-50 for Faster R-CNN [61] and Mask R-CNN [32] using our pre-trained model. All Faster/Mask R-CNN models are with the C4-backbone. We fine-tune the model end-to-end in the target datasets with a searched learning rate and keep all other parameters the same as in Detectron2 library [79]. We use the VOC07+12 `trainval` set of 16K images for training the Faster R-CNN model for 24K iterations using a batch size of 16 across 8 GPUs. The initial learning rate is reduced by a factor of 10 after 18K and 22K iterations. We also train the model using only the VOC07 `trainval` set of 5K images with smaller iterations according to the dataset size. We report results on VOC07 `test` averaged over 5 runs. We train the Mask R-CNN model ($1\times$ schedule) on the COCO 2017 `train` split and report results on the `val` split.

**Object tracking.** We further evaluate the generalization capability of the learned representations on five video tasks, including single object tracking (SOT) [78], video object segmentation (VOS) [56], multi-object tracking (MOT) [48], multi-object tracking and segmentation (MOTS) [73] and pose tracking (PoseTrack) [1]. The datasets and metrics used for the above tasks are as follows:

| Task | SOT | VOS | MOT | MOTS | PoseTrack |
|---|---|---|---|---|---|
| **Dataset** | OTB 2015 [78] | DAVIS 2017 [56] | MOT 16 [48] | MOTS [73] | PoseTrack 2017 [1] |
| **Metrics** | AUC | $\mathcal{J}$-mean | IDF1 HOTA | IDF1 HOTA | IDF1 ID-switch (IDs) |

All evaluations are based on the platform of UniTrack [76], where no additional fine-tuning is required. UniTrack [76] consists of a single and task-agnostic appearance model, which is initialized using our pre-trained model, and multiple heads to directly address different tasks without further training.

**Details of Figure 1.** In Figure 1 of the main paper, we make a comparison of transfer learning performance on five image-based tasks and five video-based tasks. The image-based tasks include linear probing (top-1 accuracy) with 800-epoch pre-trained models (LIN), semi-supervised classification (top-1 accuracy) using 1% subset of training data (SEMI), object detection (AP) on VOC dataset (VOC) and COCO dataset (COCO), instance segmentation ($AP^{mask}$) on COCO dataset (SEG). For video-based tasks, we compute rankings in terms of AUC, $\mathcal{J}$-mean, IDF-1, IDF-1 and IDF-1 for SOT, VOS, MOT, PoseTracking, and MOTS, respectively.

# D   Additional Results

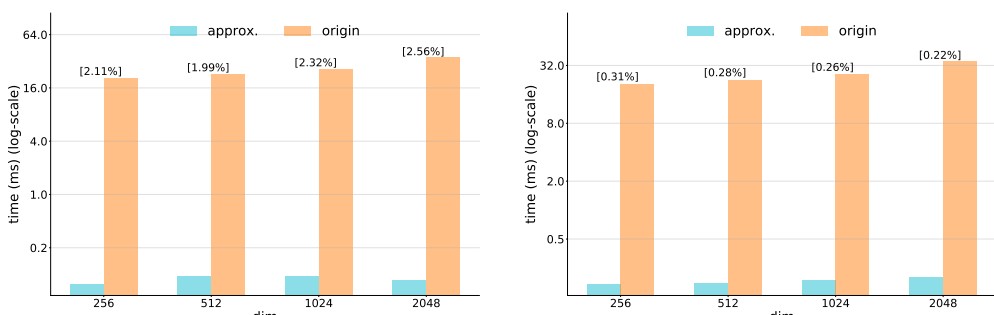

Figure 6: Comparison of running time and relative approximation error between Equation (1) (origin) and Equation (2) (approx.) for different number of samples in $Z$ (dim). Left plot: first-order approximation; Right plot: second-order approximation.

**Approximation with low-order expansion.** In Section 2.1 of the main paper, we make a comparison between original Equation (1) and our approximation using four terms of Equation (2). In this section, we provide additional comparisons using lower-order approximations. As can be seen in Figure 6, the computation process of coding length function can be even faster (*e.g.*, 0.11ms *v.s.* 35.48ms for dim 2048) with first-order approximation. But we also notice that the relative approximation error increases to 2.56%, while it is 0.22% and 0.07% for second-order and fourth-order approximation, respectively. Such approximation errors may account for the reason of performance drop on linear probing (Table 5e) and other downsteam tasks (Section 3.4).

Table 7: Comparison of linear probing results on ImageNet [17] with state-of-the-art methods. The results are reported in their original papers. † indicates methods using a projector network with large dimensions. ‡ indicates methods using multi-crop augmentation.

| Method | MEC‡ | MEC† | MEC | BYOL [29] | SimSiam [14] | Barlow [86]† | VICReg [5]† | DINO [11]‡ | UniGrad [66]†‡ |
|---|---|---|---|---|---|---|---|---|---|
| epoch | 800 | 800 | 800 | 1000 | 800 | 1000 | 1000 | 800 | 800 |
| accuracy | 75.5 | 75.1 | 74.5 | 74.3 | 71.3 | 73.2 | 73.2 | 75.3 | 75.5 |

**Pre-training with additional strategies.** In Section 3.2 of the main paper, we make a comparison of linear probing results with different methods. And in Table 1, each method is pre-trained with two $224 \times 224$ views for a fair comparison. We notice that there are multiple other strategies for self-supervised pre-training, so we provide additional experiment results in Table 7 by incorporating two widely used strategies, *i.e.*, multi-crop augmentation [10, 11] and large projector network [86, 5, 66]. Multi-crop augmentation uses additional smaller crops as local views (six $96 \times 96$ views following [10]). A larger projector network increases the dimension of each MLP layer (from 2048 to 8192 following [86, 5, 66]). We find these strategies can steadily improve the performance of MEC at the cost of more computation overhead.

**Transfer learning across different image domains.** In the experiments (Section 3) of the main paper, we have evaluated the transfer learning performance on a wide variety of image- and video-

Table 8: Transfer learning to other classification tasks with ImageNet [17] pre-trained model. The top-2 model for each datatset is **bolded** and the best model is underlined.

| Method | Food101 | CIFAR10 | CIFAR100 | SUN397 | Cars | Aircraft | VOC2007 | DTD | Pets | Caltech-101 | Flowers |
|---|---|---|---|---|---|---|---|---|---|---|---|
| *Linear probing:* | | | | | | | | | | | |
| Supervised [12] | 72.3 | **93.6** | 78.3 | 61.9 | 66.7 | **61.0** | **82.8** | 74.9 | **91.5** | **94.5** | 94.7 |
| SimCLR [12] | 68.4 | 90.6 | 71.6 | 58.8 | 50.3 | 50.3 | 80.5 | 74.5 | 83.6 | 90.3 | 91.2 |
| BYOL [29] | **75.3** | 91.3 | **78.4** | 62.2 | **67.8** | 60.6 | 82.5 | **75.5** | 90.4 | 94.2 | **96.1** |
| MEC | **75.6** | 92.1 | **78.4** | **62.7** | 67.2 | **61.5** | 82.7 | **75.8** | 90.9 | **94.6** | 96.0 |
| *Fine-tuned:* | | | | | | | | | | | |
| Random init [12] | 86.9 | 95.9 | 80.2 | 53.6 | 91.4 | 85.9 | 67.3 | 64.8 | 81.5 | 72.6 | 92.0 |
| Supervised [12] | 88.3 | 97.5 | **86.4** | **64.3** | **92.1** | 86.0 | 85.0 | 74.6 | **92.1** | 93.3 | **97.6** |
| SimCLR [12] | 88.2 | 97.7 | 85.9 | 63.5 | 91.3 | **88.1** | 84.1 | 73.2 | 89.2 | 92.1 | 97.0 |
| BYOL [29] | **88.5** | **97.8** | 86.1 | 63.7 | 91.6 | **88.1** | 85.4 | **76.2** | 91.7 | **93.8** | 97.0 |
| MEC | **88.9** | **97.8** | **86.8** | 63.8 | 91.6 | **88.5** | **85.9** | 76.0 | 91.9 | **94.9** | 97.2 |

based downstream tasks. In this section, to further evaluate whether the learned representations can generalize across different image domains, we transfer the pre-trained model to other classification tasks by linear probing and fine-tuning on 11 datasets. The results in Table 8 demonstrate that MEC's representation is more generalizable and less biased compared to the supervised baseline and other models pre-trained with specific pretext tasks, consistent with the observations in the main paper.

Table 9: Pre-training on Places365 and linear evaluation on Places365 and ImageNet dataset.

| Method | Places365 | ImageNet |
|---|---|---|
| SimCLR [12] | 53.0 | 56.5 |
| BYOL [29] | 53.2 | 58.5 |
| MEC | 53.8 | 59.9 |

**Pre-training with different datasets.** We perform self-supervised pre-training on Places365 [89] dataset using the proposed method, and then linear evaluation is conducted on Places365 and ImageNet dataset. We list the experiment results in Table 9. These results show that MEC can still learn good representations when pre-trained on different kinds of datasets, and also achieve better performance than previous methods.

## E   More Detailed Proofs

**Proof of Equation** (2). First, we rewrite Equation (1) by substituting $\mu = \frac{m+d}{2}$ and $\lambda = \frac{d}{m\epsilon^2}$, and then we can obtain the following simplified equation,

$$L = \mu \log \det \left( \boldsymbol{I_m} + \lambda \boldsymbol{Z}^\top \boldsymbol{Z} \right). \tag{5}$$

Next, we utilize the following identical equation [35],

$$\det(\exp(\boldsymbol{A})) = \exp(\mathrm{Tr}(\boldsymbol{A})), \tag{6}$$

and then we take logarithm of the both side of the above equation, which gives,

$$\log \det(\exp(\boldsymbol{A})) = \mathrm{Tr}(\boldsymbol{A}). \tag{7}$$

Let $\boldsymbol{A} = \log \left( \boldsymbol{I_m} + \lambda \boldsymbol{Z}^\top \boldsymbol{Z} \right)$, then we have,

$$\log \det \left( \boldsymbol{I_m} + \lambda \boldsymbol{Z}^\top \boldsymbol{Z} \right) = \mathrm{Tr} \left( \log \left( \boldsymbol{I_m} + \lambda \boldsymbol{Z}^\top \boldsymbol{Z} \right) \right). \tag{8}$$

So Equation (5) can be reformulated as,

$$\begin{aligned} L &= \mu \log \det \left( \boldsymbol{I_m} + \lambda \boldsymbol{Z}^\top \boldsymbol{Z} \right) \\ &= \mathrm{Tr} \left( \mu \log \left( \boldsymbol{I_m} + \lambda \boldsymbol{Z}^\top \boldsymbol{Z} \right) \right). \end{aligned} \tag{9}$$

Finally, we apply Taylor series expansion to expand the logarithm of the matrix in the above equation, and obtain Equation (2),

$$L = \mathrm{Tr}\left( \mu \sum_{k=1}^{\infty} \frac{(-1)^{k+1}}{k} \left( \lambda \boldsymbol{Z}^\top \boldsymbol{Z} \right)^k \right), \tag{10}$$

with convergence condition: $\left\| \lambda \boldsymbol{Z}^\top \boldsymbol{Z} \right\|_2 < 1$.

**Proof of convergence condition of Equation (3).** To ensure the convergence of Equation (3), we require,

$$\| \boldsymbol{C} \|_2 < 1, \tag{11}$$

where $\boldsymbol{C} = \lambda \boldsymbol{Z}_1^\top \boldsymbol{Z}_2$ and $\lambda = \frac{d}{m\epsilon^2} = \frac{1}{m\epsilon_d^2}$. We note the inequality between matrix norms,

$$\| \boldsymbol{C} \|_2 \le \sqrt{\| \boldsymbol{C} \|_1 \| \boldsymbol{C} \|_\infty}, \tag{12}$$

which is a special case of Hölder's inequality. Since 1-norm of matrix is simply the maximum absolute column sum of the matrix, we have

$$\| \boldsymbol{C} \|_1 = \max_{1 \le j \le m} \sum_{i=1}^{m} |c_{ij}|. \tag{13}$$

Note that the columns of $\boldsymbol{Z}_1$ and $\boldsymbol{Z}_2$ are $\ell_2$-normalized embeddings, so we have,

$$\| \boldsymbol{C} \|_1 = \max_{1 \le j \le m} \sum_{i=1}^{m} |c_{ij}| \le \lambda m. \tag{14}$$

Similarly, we can obtain,

$$\| \boldsymbol{C} \|_\infty = \max_{1 \le i \le m} \sum_{j=1}^{m} |c_{ij}| \le \lambda m. \tag{15}$$

Finally, we go back to Equation (12) and obtain,

$$\| \boldsymbol{C} \|_2 \le \sqrt{\| \boldsymbol{C} \|_1 \| \boldsymbol{C} \|_\infty} \le \lambda m. \tag{16}$$

To ensure that the convergence condition of Equation (11) can be strictly satisfied, we require,

$$\lambda < \frac{1}{m}, \tag{17}$$

or we can equally set $\epsilon_d^2 > 1$ by adjusting the degree of distortion. Note that in the Section 2.2 of the main paper, we simply set $\epsilon_d^2 = \frac{d}{m}$ for $d > m$, which is the case of the preliminary experiments on CIFAR-10 [41] (the feature dimension $d = 2048$ and the batch size $m = 1024$). In practice, we empirically find that the Taylor expansion converges over a wide range of $\epsilon_d$ (see Figure 4(c) and Table 5c).

**Proof of Equation (4).** Equation (4) is a direct result of Sylvester's determinant identity, which states that if $A$ and $B$ are matrices of sizes $m \times d$ and $d \times m$, then we have,

$$\det\left( I_m + AB \right) = \det\left( I_d + BA \right), \tag{18}$$

and it can be proved by the following derivation,

$$
\begin{aligned}
&\det \begin{pmatrix} I & -B \\ A & I \end{pmatrix} \det \begin{pmatrix} I & B \\ 0 & I \end{pmatrix} \\
&= \det \begin{pmatrix} I & -B \\ A & I \end{pmatrix} \begin{pmatrix} I & B \\ 0 & I \end{pmatrix} = \det \begin{pmatrix} I & 0 \\ A & AB + I \end{pmatrix} = \det(I_m + AB),
\end{aligned} \tag{19}
$$

and we also have,

$$
\begin{aligned}
&\det \begin{pmatrix} I & B \\ 0 & I \end{pmatrix} \det \begin{pmatrix} I & -B \\ A & I \end{pmatrix} \\
&= \det \begin{pmatrix} I & B \\ 0 & I \end{pmatrix} \begin{pmatrix} I & -B \\ A & I \end{pmatrix} = \det \begin{pmatrix} I + BA & 0 \\ A & I \end{pmatrix} = \det(I_d + BA).
\end{aligned} \tag{20}
$$

So Equation (18) is proved. Let $A = \lambda \boldsymbol{Z}_1^\top$ and $B = \boldsymbol{Z}_2$, and using the fact that the determinant of the transpose of a square matrix is equal to the determinant of the matrix, then we can prove Equation (4),

$$\mathcal{L}_{MEC} = \underbrace{-\mu \log \det\left(\boldsymbol{I_m} + \lambda \boldsymbol{Z}_1^\top \boldsymbol{Z}_2\right)}_{\text{batch-wise}} = \underbrace{-\mu \log \det\left(\boldsymbol{I_d} + \lambda \boldsymbol{Z}_1 \boldsymbol{Z}_2^\top\right)}_{\text{feature-wise}}. \tag{21}$$

**Relation to SimSiam [14] and BYOL [29].** SimSiam [14] uses negative cosine similarity as loss function. And it is equivalent to the mean squared error of $\ell_2$-normalized vectors, up to a scale of 2, which is the loss used in BYOL [29]. We write the loss function of SimSiam [14] as the following equation,

$$\mathcal{L}_{SimSiam} = -\sum_{i=1}^{m} z_1^i \cdot z_2^i, \tag{22}$$

where $z_1^i$ and $z_2^i$ are the embeddings of two views of the same image $i$. By taking Taylor expansion (Equation (2)) of the *left* side of Equation (4), we obtain,

$$\mathcal{L}_{MEC}^{n=1} = - \operatorname{Tr}\left(\mu \lambda \boldsymbol{Z}_1^\top \boldsymbol{Z}_2\right) = -\mu \lambda \sum_{i=1}^{m} z_1^i \cdot z_2^i, \tag{23}$$

which is equivalent to Equation (22) up to a scale of $\mu\lambda$. Since $\mu\lambda$ is a constant and can be absorbed by adjusting the learning rate during optimization, the objective function of SimSiam [14] and BYOL [29] can be viewed as the first order expansion of the objective function of MEC.

**Relation to Barlow Twins [86] and VICReg [5].** Barlow twins [86] aims to make the cross-correlation matrix computed from twin embeddings as close to the identity matrix as possible, with an invariance term and a redundancy reduction term. VICReg [5] follows the similar idea by using a term that decorrelates each pair of variables along each branch of Siamese networks. The objective function of Barlow twins [86] is as follows,

$$\mathcal{L}_{Barlow} = \sum_{i=1}^{d} \left(1 - \boldsymbol{C}_{ii}\right)^2 + \lambda_{barlow} \sum_{i=1}^{d} \sum_{j \neq i}^{d} \boldsymbol{C}_{ij}^2, \tag{24}$$

where $\boldsymbol{C}$ is the feature-wise cross-correlation matrix and $\lambda_{barlow}$ is a positve constant. By taking Taylor expansion (Equation (2)) of the *right* side of Equation (4), we obtain,

$$\begin{aligned}
\mathcal{L}_{MEC}^{n=2} &= - \operatorname{Tr}\left(\mu \lambda \boldsymbol{Z}_1 \boldsymbol{Z}_2^\top - \frac{\mu}{2}\left(\lambda \boldsymbol{Z}_1 \boldsymbol{Z}_2^\top\right)^2\right) \\
&= \mu \sum_{i=1}^{d} \left(-\boldsymbol{C}_{ii} + \frac{1}{2}\boldsymbol{C}_{ii}^2\right) + \frac{\mu}{2} \sum_{i=1}^{d} \sum_{j \neq i}^{d} \boldsymbol{C}_{ij}^2,
\end{aligned} \tag{25}$$

where $\boldsymbol{C} = \lambda \boldsymbol{Z}_1 \boldsymbol{Z}_2^\top$. And the above equations show that the objective function of Equation (24) can be viewed as the second-order expansion of the objective function of MEC. We notice that Barlow twins [86] uses batch normalization rather than $\ell_2$ normalization on the embeddings $z$, and we show that these two kinds of normalization techniques have similar effects on both Barlow twins [86] and our MEC:

| method | batch norm | $\ell_2$ norm | linear |
|---|:---:|:---:|---|
| Barlow Twins [86] | ✓ | | 67.3 |
| | | ✓ | 67.4 |
| MEC | ✓ | | 70.6 |
| | | ✓ | 70.6 |

**Relation to SimCLR [12] and MoCo [13].** SimCLR [12] and MoCo [13] are two typical contrastive learning methods that aim to push negative pairs apart while pulling positive pairs together. And they

use the following InfoNCE loss function [53],

$$\mathcal{L}_{InfoNCE} = -\sum_{i=1}^{m}(c_{i,i}/\tau) + \sum_{i=1}^{m}\log\sum_{j\neq i}^{m}(\exp(c_{i,j}/\tau) + \exp(c_{i,i}/\tau)) \qquad (26)$$

where $\tau$ is the temperature parameter. By taking Taylor expansion (Equation (2)) of the *left* side of Equation (4), we obtain,

$$\mathcal{L}_{MEC}^{n=2} = -\operatorname{Tr}\left(\mu\lambda \mathbf{Z}_1^\top \mathbf{Z}_2 - \frac{\mu}{2}\left(\lambda \mathbf{Z}_1^\top \mathbf{Z}_2\right)^2\right) \qquad (27)$$

We notice that although the above two equations do not take the exactly same forms, they have similar effects on the learning process: the first term aims to model the invariance with respect to data augmentations; and the second term aims to minimize the similarity between negative samples.

## F   Limitations and Future Work

The estimation of entropy from a finite set of high-dimensional vectors is itself a challenging problem in the field of statistical learning. So in this work, as an exploration of the principle of maximum entropy in self-supervised learning, we opt to exploit a computationally tractable surrogate for the entropy of representations. In order to facilitate large-scale pre-training, we further leverage Taylor series expansion to accelerate the computation process and we notice that more theoretical investigation is needed for the empirical convergence of small distortion. Future work can seek a more direct entropy estimator for maximum entropy coding. The proposed method aims to alleviate the bias introduced by the specific pretext task, and we notice that the bias can also be introduced by the designed data augmentations, which is a common problem in current self-supervised learning methods. Future work may seek automated data augmentation strategies and further generalize the proposed method to other modalities (*e.g.*, audio, text).

## G   Broader Impact

As the first method that introduces the principle of maximum entropy into self-supervised learning, the presented work may inspire more methods leveraging this principle towards learning generalizable representations, which is the core of self-supervised learning. As we demonstrate in the experiments, the proposed method positively contributes to a wide variety of vision tasks, such as image classification, object detection and tracking. However, there is also a potential that the proposed method has negative societal impacts for a particular use of the applications. The proposed method learns representations from large-scale datatsets and the learned representations may reflect the data biases inherent in the datasets.

## H   Licenses of Assets

CIFAR-10 [41] is subject to MIT license. VOC [25] data includes images obtained from the Flickr website and use of these images is subject to the Flickr terms of use. COCO [44] is subject to the Creative Commons Attribution 4.0 License. ImageNet [17] is subject to the licenses on the website[2].

---

[2]https://www.image-net.org/download