# OpenReview forum: "Self-Supervised Learning via Maximum Entropy Coding"
_NeurIPS.cc/2022/Conference — NeurIPS 2022 Accept_

### Official Review · Reviewer_CgUD · 2022-06-21

**Rating:** 8
**Confidence:** 4
**Soundness:** 4 excellent
**Presentation:** 4 excellent
**Contribution:** 3 good

**Summary:**

This paper proposes a self-supervised learning method dubbed Maximum Entropy Encoding (MEC) which leverages the principle of maximum entropy to learn unbiased representations of an image dataset (experiments done on ImageNet). The authors combine a maximum entropy with the augmentation-invariance objective of contrastive learning, which is justified as a view consistency prior which gives testable information.
The exact resulting loss has a log determinant, which they approximate using a Taylor series. They show that various orders of this approximation are in fact equivalent to existing self-supervised methods such as SimSiam and Barlow Twins.
Experiments are conducted with pretraining on ImageNet and suitably diverse downstream tasks: linear evaluation on ImageNet, semisupervised classification on ImageNEt, transfer learning on object detection (VOC + COCO) and segmentation (COCO), as well as video tracking (Table 5).

**Questions:**

From Strengths-Quality: Can you clarify where the increase in metrics from cited sources comes from? Was I looking in the wrong place in the cited works or are there improvements stemming from more modern implementations?

From Weaknesses-Quality:
Are you able to provide any results with pretraining on moderate-to-large scale datasets besides ImageNet?
I'd also be curious if this objective can provide improvements for self-supervised learning on smaller-scale datasets (this latter point is an extra ask, if pressed for space in the rebuttal feel free to ignore)

**Limitations:**

Appendices E and F provide a succinct but accurate and precise description of limitations and the possible relation to societal impact

**Strengths And Weaknesses:**

Strengths:
Originality:
The community has been iteratively constructing contrastive or similar self-supervised learning methods for several years now. This paper subsumes several previous works by providing a novel unified view that allows for further adaption and exploration. While the resulting loss code is not markedly different from existing work, the theoretical approach it is derived from and the resulting flexibility/opportunity it yields constitute originality in my eyes.

Quality:
The experiments in this paper are very clean, with straightforward well-accepted experimental settings. Appendix C describes the methodology well without introducing any complicating bells and whistles. The results of the new method are consistently equal or superior to existing objectives all while being very motivated.
I cross-referenced numbers with original works and the authors match (or in some cases exceed) the original citations in table referenced numbers (example: the arxiv version of SimCLR lists 69.3 as its top1 imagenet acc while this work gives up to 70.4). I assume these discrepancies are from improved/additional facets in the training, but would appreciate clarification on this point.
I feel extremely comfortable that I could replicate the experiments of this paper with minimal effort and would see comparable results.

Clarity:
This paper is extremely well-written. I was able to understand all of the concepts on the first read-through. The order of presentation was logical and descriptions were enlightening without being overly wordy. As previously mentioned, I feel comfortable that I could replicate the results quickly, especially given the useful pseudocode in Appendix A.

Significance:
The direct tying of a family of objectives to a very grounded mathematical concept is highly significant in my eyes as it brings the iterative line of research to a convergence point from which more novel lines of research can be taken. There could have been several more iterations of self-supervised learning papers converging to this objective by empirical motivation but this work shortcuts that and opens up exciting possibilities for future work (especially given the prominence of contrastive learning objectives in current vision-language works).

Misc:
- Figure 4 was an extremely useful illustration
- Appendix was extremely informative and helpful
- The ablations in Table 5 (particularly (e)) largely match my intuition which is nice

Weaknesses:
Originality:
This paper consolidates existing lines of exploration and opens the door for further "contrastive" methods and as such the ultimate objective is not technically very different from existing work. Given the importance and significance of the consolidation though, this is not truly a weakness in my opinion.

Quality:
*I do wish that there were at least some pretraining experiments on datasets besides ImageNet (either uncurated web imagery or something like Places365). In my opinion this is the biggest weakness of this paper.*
As stated above, overall I was very impressed with the quality of this work. The only typo that jumped out to me while reading was "variaty" instead of "variety" in L60.

Clarity:
I am very familiar with the contrastive learning so am probably a biased estimator here, but I thought the explanations and experiments were extremely straightforward and non-confusing. I have no complaints about the clarity. My only related suggestion is that a pointer to Appendix H is placed in the main text (see Misc below)

Significance:
As stated in the above originality section, it's difficult for a consolidation paper to have earth-shattering impact as the value largely lies in the tying together of work that necessarily exists. I don't think that this by any means is a net negative; I would rank this paper as one of the more significant ones that I've come across in the last 2-4 weeks of arxiv, but it does cap it below what a Conference Best Paper might look like.


Misc:
- Appendix H (starting from L724) is extremely interesting and I was wishing for a similar section within the main text. While incorporating that many lines is impractical at this stage of the paper; I would suggest that a reference to Appendix H with a preview of what it contains is added to the main text

---

> ### Author Response · Authors · 2022-08-02
> **Response to Reviewer CgUD**
>
> Thank you very much for your constructive comments and support. We greatly appreciate that you found our work being very motivated and opening up exciting possibilities for future work. Below we address the raised concerns.
>
> > **1. "Can you clarify where the increase in metrics from cited sources comes from?"**
>
> The comparison of different methods in Tab.1 is based on the reproduction results from the SimSiam paper [A]. In order for a fair comparison, they made small and straightforward modifications to the related methods. And the reproduction has better results for SimCLR, MoCo v2, and SwAV, and has comparable results for BYOL. Please kindly refer to Appendix C of the SimSiam paper [A] for more details. We will add a note in the paper to better clarify this.
>
> > **2. "Are you able to provide any results with pretraining on moderate-to-large scale datasets besides ImageNet? I'd also be curious if this objective can provide improvements for self-supervised learning on smaller-scale datasets."**
>
> Thank you for your insightful questions. We perform self-supervised pre-training on Places365 dataset using the proposed method, and then linear evaluation is conducted on Places365 and ImageNet dataset. We list the initial experiment results in the table below.
>
> | Method | Places365 | ImageNet |
> |--------|-----------|----------|
> | SimCLR | 53.0      | 56.5     |
> | BYOL   | 53.2      | 58.5     |
> | MEC    | **53.8**  | **59.9** |
>
> These results show that MEC can still learn good representations when pre-trained on different kinds of datasets, and also achieve better performance than previous methods. As for pre-training on smaller-scale datasets, we show in the main paper that MEC improves both linear and kNN accuracy of other methods by a large margin (e.g., 2.3\% linear probing accuracy for SimCLR) on CIFAR-10 dataset when working as a regulation term (Fig.5), and when working as a standalone pre-training objective, MEC gains +2.9\% accuracy over SimCLR evaluated by linear probing. We will include these results and discussions in the paper to better demonstrate the effectiveness of MEC.
>
> > **3. "'variaty' instead of 'variety' in L60" "Appendix H (starting from L724) is extremely interesting and I was wishing for a similar section within the main text."**
>
> Thank you for your very helpful suggestions. We will revise the paper accordingly, and add a reference to Appendix H with a preview of what it contains in the main text.
>
> **References**
>
> [A] Chen, Xinlei, and Kaiming He. "Exploring simple siamese representation learning." CVPR 2021.

---

### Official Review · Reviewer_Z5uT · 2022-07-04

**Rating:** 5
**Confidence:** 4
**Soundness:** 2 fair
**Presentation:** 3 good
**Contribution:** 2 fair

**Summary:**

This paper proposed a joint-embedding objective called Maximum Entropy Coding, which is close to MCR^2 [74]. This method directly optimizes the information content by minimizing the coding length function. This proposed method unifies batch-wise objectives (SimSiam) and feature-wise objectives (BarlowTwins). Practically, this is implemented as a Taylor series expansion.
The proposed method shows strong performances (when combined with all existing techniques, including exponential moving average and asymmetric network) on a wide range of experiments, including ImageNet linear probe, semi-supervised classification, transfer learning on video tasks, and object detection.
However, this paper actually uses all the tricks, including exponential moving average from BYOL, but only mentions it in the ablation study/supplementary material. It is obvious that the strong result of this method mostly comes from this mixture of existing design components.

**Questions:**

Do you consider n=2 (Taylor expansion order) as a novel model? If not, then how is the performance of this baseline across all experiments. If yes, how is it different from BarlowTwins?

**Strengths And Weaknesses:**

Strengths:
1. The proposed method is strongly supported by theoretical motivation: the maximum entropy principle.
2. The proposed method shows strong performances on a wide range of experiments (when combined with all existing techniques, including exponential moving average, asymmetric network), including ImageNet linear probe, semi-supervised classification, transfer learning on video tasks, and object detection.
3. The authors provided experimental details, including pseudo-code and all hyperparameters.

Weaknesses:
1. The strong result mostly comes from a mixture of designing components, e.g., exponential moving average and asymmetric architecture. It is questionable how much advantage comes from the maximum entropy coding loss. In fact, the only contribution of the proposed method over existing methods is that it has a higher-order correction. Table 5e clearly shows that the extra two orders (2->4) only increase accuracy by 0.3%.
2. It's unclear what the main contribution is to this paper.
(a) The "principle of maximum entropy" is no different from requiring independent features, which is already proposed in MCR^2, BarlowTwins, VICReg, and whitening SSL.
(b) The higher-order approximation, as mentioned above, only has a 0.3% linear probe improvement. All other experiments fail to verify the advantage.
(c) The practical advantage of this mixture model. The authors only compare their work to fundamental frameworks (BYOL, BarlowTwins, SwAV), but there are many similar mixture models, e.g., arXiv:2204.07141, arXiv:2104.14548, arXiv:2109.12909, arXiv:2012.13493, arXiv:2201.05119

====== after rebuttal comments =====

I am more convinced by the authors that the empirical advantage is misleading. The empirical advantage is exactly just 0.3%. Actually, in response point 1, Barlow Twins are supposed to reach 73.5%. So here, the extra two orders approximately indeed only provide a 0.1% improvement.

I think this is totally ok. A good theory unifies several ideas and provides incremental improvements. But it is unacceptable that the paper tries to hide this point.

The main figure and most of the results tables will mislead readers that the advantage of the proposed method over Barlow Twins / SimCLR is due to the proposed loss. In fact, leveraging other tricks like momentum encoders is the main reason but is only mentioned in L159 once. Not in the figure, not in the main text, not in the experimental detail.

---

> ### Author Response · Authors · 2022-08-02
> **Response to Reviewer Z5uT (Part 1)**
>
> We thank the reviewer for the valuable comments, and for recognizing the strong performances and theoretical motivation of our work. We address the raised concerns below.
>
> > **1. "The strong result mostly comes from a mixture of designing components, e.g., exponential moving average and asymmetric architecture. It is questionable how much advantage comes from the maximum entropy coding loss."**
>
> Although the mixture of designing components contributes to the results (Tab.5), the maximum entropy coding loss is the most important reason for the strong performances of our method. When removing all those components, our method still outperforms SimCLR, Barlow Twins by 3.2, 0.6 points, respectively. And this behavior is different from other SSL methods (e.g., SimSiam, BYOL), where mode collapse is a big concern without those components. We summarize the results in the table below.
>
> | Method       | Ema | Asym. | Linear   |
> |--------------|-----|-------|----------|
> | SimCLR       | no  | no    | 70.4     |
> | BYOL         | no  | no    | collapse |
> | SimSiam      | no  | no    | collapse |
> | Barlow Twins | no  | no    | 73.0     |
> | MEC          | no  | no    | **73.6** |
>
> The technique of exponential moving average has been common practice in recent SSL methods to improve performance. However, using asymmetric architecture could decrease the performance, sometimes by a large margin (61.3 v.s. 71.4, as reported in Barlow Twins [A]). So a mixture of designing components does not necessarily lead to performance improvement.
>
> To further validate how much advantage comes from the maximum entropy coding loss, below we compare our method with BYOL on a wide variety of downstream tasks.
>
> | Method   | order | Ema | Asym. | Linear   | Semi     | Det      | Ins      | SOT      | VOS      | MOT      | MOTS     | PoseTrack |
> |----------|-------|-----|-------|----------|----------|----------|----------|----------|----------|----------|----------|-----------|
> | BYOL [B] | n=1   | yes | yes   | 74.3     | 53.2     | 37.9     | 33.2     | 58.9     | 58.8     | 62.9     | 70.8     | 73.8      |
> | MEC      | n=4   | yes | yes   | **74.5** | **55.9** | **39.8** | **34.7** | **62.3** | **62.0** | **63.3** | **72.0** | **74.1**  |
>
> Although both methods use exponential moving average and asymmetric architecture, MEC outperforms BYOL on all 9 tasks considered. Similar trends can also be observed on 11 fine-grained classification benchmarks (Tab.8). These experiment results further emphasize the importance of the maximum entropy coding loss instead of the mixture of designing components.
>
> > **2. "The higher-order approximation, as mentioned above, only has a 0.3\% linear probe improvement. All other experiments fail to verify the advantage."**
>
> As the main purpose of our method is to improve the generalization of SSL representation on various downstream tasks, we argue that the advantage should not be solely evaluated by ImageNet linear probe. Instead, we compare transferred performance on a variaty of downstream tasks.
>
> As can be seen from the table in response 1, with fourth-order approximation, MEC outperforms BYOL (which can be seen as MEC with first-order approximation) on all 9  tasks considered across the board, suggesting the advantage of using higher-order approximation. We also note that the extra two orders (2->4) benefit less than lifting from the first order to the fourth order approximation. It is reasonable since the relative approximation error of second-order expansion is already quite decent, lower than 0.5\% (Fig.6), almost the same as fourth-order expansion (Fig.3). We will include these discussions in the paper to better clarify the advantage of using higher-order approximation.

---

> > ### Author Response · Authors · 2022-08-02
> > **Response to Reviewer Z5uT (Part 2)**
> >
> > > **3. "The 'principle of maximum entropy' is no different from requiring independent features, which is already proposed in MCR$^{2}$, BarlowTwins, VICReg, and whitening SSL"**
> >
> > Since they can be categorized as feature-wise SSL methods, these methods and ours indeed require each feature dimension to be independent. However, what really matters is the different ways of learning independent features proposed in these methods. In contrast to other mentioned methods, our method aims to learn general-purpose representations and is strongly supported by theoretical motivation. In addition, our proposed method unifies the batch-wise and feature-wise objectives as low-order approximations of our method, and this new perspective also distinguishes our method from existing ones.
> >
> > > **4. "Do you consider n=2 (Taylor expansion order) as a novel model?"**
> >
> > Yes, we do consider the second-order approximation as a novel model. Despite that the second-order expansion of the feature-wise side of Eqn.4 is equivalent to Barlow Twins, our novelty comes from two aspects.
> >
> > First, the motivation and derivation are novel. Barlow Twins aims to minimize the redundancy between the feature components to avoid mode collapse, while our derivation originates from the desire of optimizing generalization on downstream tasks by leveraging the principle of maximum entropy.
> >
> > Second, the perspective we provided to the community is novel. We find interesting equivalence between low-order approximations of MEC and existing batch-wise or feature-wise objectives, which provides a new perspective for a unified understanding of prevalent SSL methods. And as remarked by Reviewer CgUD, "The direct tying of a family of objectives to a very grounded mathematical concept is highly significant in my eyes as it brings the iterative line of research to a convergence point from which more novel lines of research can be taken."
> >
> > > **5. "The authors only compare their work to fundamental frameworks (BYOL, BarlowTwins, SwAV), but there are many similar mixture models, e.g., arXiv:2204.07141, arXiv:2104.14548, arXiv:2109.12909, arXiv:2012.13493, arXiv:2201.05119"**
> >
> > Thanks for pointing out these relevant and interesting papers. We will include all these papers in the related work section, and compare our work to these methods (see the preliminary table below) if their code and pre-trained models are available, so we can reproduce the missing results and fit them in our main tables (Tab.1-4).
> >
> > | Method       | Linear100 | Linear200 | Linear300 | Det  | Ins  |
> > |--------------|-----------|-----------|-----------|------|------|
> > | NNCLR [C]    | 69.4      | 70.7      | -         | -    | -    |
> > | HEXA [D]     | -         | 68.9      | -         | -    | -    |
> > | C-SimCLR [E] | -         | -         | 70.1      | -    | -    |
> > | C-BYOL [E]   | -         | -         | 73.6      | -    | -    |
> > | MEC          | 70.6      | 71.9      | -         | 39.8 | 34.7 |
> >
> > **References**
> >
> > [A] Zbontar, Jure, et al. "Barlow twins: Self-supervised learning via redundancy reduction." ICML 2021.
> >
> > [B] Grill, Jean-Bastien, et al. "Bootstrap your own latent-a new approach to self-supervised learning." NeurIPS 2020.
> >
> > [C] Dwibedi, Debidatta, et al. "With a Little Help from My Friends: Nearest-Neighbor Contrastive Learning of Visual Representations." arXiv:2104.14548 (2021).
> >
> > [D] Li, Chunyuan, et al. "Self-supervised pre-training with hard examples improves visual representations." arXiv:2012.13493 (2020).
> >
> > [E] Lee, Kuang-Huei, et al. "Compressive Visual Representations." arXiv:2109.12909 (2021).

---

> > > ### Comment · Reviewer_Z5uT · 2022-08-06
> > > **Thanks for the response**
> > >
> > > I thank the authors for the detailed response.
> > > This is a very well-written paper with solid theoretical and empirical support.
> > >
> > > However, I am more convinced by the authors that the empirical advantage is misleading. The empirical advantage is exactly just 0.3%.
> > > Actually, in response point 1, Barlow Twins are supposed to reach 73.5%. So here, the extra two orders approximately indeed only provide a 0.1% improvement.
> > >
> > > I think this is totally ok. A good theory unifies several ideas and provides incremental improvements. But it is unacceptable that the paper tries to exaggerate the results.
> > >
> > > The main figure and most of the results tables will mislead readers that the advantage of the proposed method over Barlow Twins / SimCLR is due to the proposed loss. In fact, leveraging other tricks like momentum encoders is the main reason but is only mentioned in L159 once. Not in the figure, not in the main text, not in the experimental detail.
> > >
> > > I will lower my score due to this reason.

---

> > > > ### Author Response · Authors · 2022-08-08
> > > > **Thanks for the additional feedback**
> > > >
> > > > We sincerely thank the reviewer for providing the additional feedback. We address your concerns below.
> > > >
> > > > >**"Actually, in response point 1, Barlow Twins are supposed to reach 73.5\%. So here, the extra two orders approximately indeed only provide a 0.1\% improvement."**
> > > >
> > > > **First, it should be clarified that comparing the mentioned number (73.5\%) with ours (73.6\%) is *unfair*, because of differences in training settings.** The settings differ in two aspects: the training scheme and the projector network.
> > > > - The comparison made in response point 1 (also in Tab.1 of our paper) is based on 800-epoch pre-trained models in order for a fair comparison of all methods. The result that the reviewer mentioned is based on a 1000-epoch pre-trained model, which is the experiment setting of Barlow Twins.
> > > > - Barlow Twins uses a larger projector network, the dimension of which is 8192.  In contrast, we only use a 2048-d projector network in order for a fair comparison of other methods (e.g., SimCLR, MoCo v2, SimSiam). Even so, our method outperforms Barlow twins by 0.6\% evaluated by linear probing.The ablation experiments in Barlow Twins show their performance drops by over 3\% when reducing the dimension from 8192 to 2048. However, this result is obtained with a 300-epoch training scheme.  When trained for 800 epochs, we did not find a relevant result, but it is natural to expect a similar performance drop.
> > > >
> > > > A detailed comparison with consideration of experimental settings is shown below. As summary, when experimental settings are controlled to be identical for a fair comparison, **the performance gap between MEC and Barlow Twins is far more than 0.1\%.**
> > > >
> > > > | method       | source         | pre-training epochs | projector width | linear evaluation |
> > > > |--------------|----------------|:---------------------:|:-----------------:|:-------------------:|
> > > > | Barlow Twins | github release | 1000                | 8192            | 73.5              |
> > > > | Barlow Twins | original paper | 1000                | 8192            | 73.2              |
> > > > | Barlow Twins | reproduction   | 800                 | 8192            | 73.0              |
> > > > | Barlow Twins | estimated | 800                 | 2048            | <73.0              |
> > > > | MEC          | ours           | 800                 | 2048            | 73.6              |
> > > >
> > > >
> > > > **Second, we would like to emphasize again that the main purpose of our method is to improve the *generalization of SSL representation on various downstream tasks***, therefore we do not recommend being obsessed solely with the ImageNet linear probe results. Comparing MEC with Barlow Twins, the former outperforms the latter on all tasks across the board, including semi-supervised learning, object detection, instance segmentation and object tracking. It is important to point out that results in Figure 1, Table 1-4 together make our point, not Table 1 alone.
> > > >
> > > > >**"But it is unacceptable that the paper tries to exaggerate the results. [...] Not in the figure, not in the main text, not in the experimental detail."**
> > > >
> > > > We respectfully disagree with this point. First, in the method section (L158-160), we use a separate paragraph to clearly state that the additional techniques (momentum encoder and asymmetric networks) can improve the performance of the minimalist variant of MEC (shown in Fig.2).
> > > >
> > > > Second, in the experiments section (Tab.5, L252-259),  we use six tables to show the default settings of our experiments, and the results demonstrate the effects of different designs of Siamese networks and also the advantage of adopting momentum encoder and asymmetric networks. Moreover, we list all the implementation details of the network architecture and momentum encoder in Appendix C (L586-595, L599-600).
> > > >
> > > > Third, we show in part 1 of the response that our MEC can outperform other methods on various downstream tasks with or without the additional techniques, which emphasizes the importance of the proposed loss instead of the architecture design or momentum encoder. We will revise the paper accordingly to better clarify the gains of different parts of our method.
> > > >
> > > > -----
> > > >
> > > > We hope to have addressed your concerns. Please let us know if you have any further suggestions or concerns.

---

### Official Review · Reviewer_QNyH · 2022-07-10

**Rating:** 6
**Confidence:** 5
**Soundness:** 3 good
**Presentation:** 3 good
**Contribution:** 3 good

**Summary:**

This paper aims to learn generalizable representations without labels. To this end, this paper proposes Maximum Entropy Coding (MEC), inspired by the principle of maximum entropy in information theory. MEC uses the minimal lossy coding and the Taylor series approximation to make the maximum entropy estimation feasible. Extensive experimental results show that MEC consistently outperforms existing SSL methods under various downstream tasks. Furthermore, this paper demonstrates that MEC is robust to various hyperparameters (e.g., smaller batch sizes) and architectures (e.g., ViTs).


**Questions:**

- Too small top margins in p2-3. **This seems to violate formatting rules.** This should be modified in the rebuttal period. Also, other spaces (e.g., caption spaces) seem too narrow.
- Could you test more transfer learning experiments with fine-grained classification benchmarks? These benchmarks are also important to evaluate the transferability of the learned representations. For the transfer setup, I recommend seeing SimCLR and BYOL papers.


**Limitations:**

This paper well addressed the limitations and the potential negative societal impact.


**Strengths And Weaknesses:**

The proposed method has several strengths: (1) simplicity and scalability, (2) high performance across various downstream tasks, and (3) robustness to various hyperparameters and architectures. I think the extensive experiments well demonstrate these strengths. Furthermore, MEC can be interpreted as batch-wise and feature-wise SSL methods, so I feel this part is also interesting.

One major concern with this paper is that what information the proposed method can learn while existing SSL methods cannot is unclear. This paper mentioned that existing SSL methods introduce some "biases" while MEC does not. What are the biases exactly? Could you provide some empirical evidence supporting that MEC can learn less-biased representations? I think this paper does not explain why and how MEC can outperform other SSL methods.

Some minor concerns are provided in the **Questions** section.

To sum up, although some explanation seems insufficient, I feel the empirical results are strong. Hence, I vote for Weak Accept.

---

> ### Author Response · Authors · 2022-08-02
> **Response to Reviewer QNyH (Part 1)**
>
> We thank the reviewer for the positive comments and constructive feedback. Below we address the raised concerns.
>
> > **1. "What information the proposed method can learn while existing SSL methods cannot is unclear.""What are the biases exactly? Could you provide some empirical evidence supporting that MEC can learn less-biased representations?"**
>
> Thank you for the insightful questions. The bias of representations is a tendency to prefer some particular aspects of data over others. In self-supervised learning, the representations are learned through solving pretext tasks, and hence the bias is closely related to the nature of different pretext tasks.
>
> - Early approaches, such as image colorization and orientation prediction, bias the model to learn low-level image statistics, instead of high-level visual concepts, thus leading to poor empirical performance on high-level downstream tasks.
>
> - A most prevalent line of self-supervised learning methods, i.e. contrastive learning, bases themselves on the instance discrimination pretext task.  However, such learned representations are found biased to image-level tasks such as image classification, while by contrast degenerate in patch- or pixel-level tasks like object detection and semantic segmentation [C].
>
> - Some pretext tasks are designed tailored to given downstream applications. For instance, DetCo [C] is designed specifically for the object detection task, thus  the learned representation naturally  degenerates on other tasks like ImageNet classification.
>
> Considering that the ultimate goal of self-supervised learning is a general-purpose representation, we emphasize that our contribution which distinguishes our method from existing ones is that we *explicitly* encourage the *generalization ability on downstream tasks*  and *minimize the bias* in the formulation of the pretext task, by introducing the Maximum Entropy Principle. The empirical results show that MEC generalizes well consistently across (i) image-based and video-based tasks (Tab.1,2,4), (ii) patch-level and pixel-level tasks (Tab.3). We summarize the results in the table below.
>
> | Method          | Linear   | Semi     | Det      | Ins      | SOT      | VOS      | MOT      | MOTS     | PoseTrack |
> |-----------------|----------|----------|----------|----------|----------|----------|----------|----------|-----------|
> | BYOL [B]        | 74.3     | 53.2     | 37.9     | 33.2     | 58.9     | 58.8     | 62.9     | 70.8     | 73.8      |
> | BarlowTwins [D] | 73.0     | 55.0     | 39.2     | 34.3     | 60.5     | 61.7     | 62.4     | 69.8     | **74.3**  |
> | MEC             | **74.5** | **55.9** | **39.8** | **34.7** | **62.3** | **62.0** | **63.3** | **72.0** | 74.1      |
>
> Furthermore, for a fixed given downstream task, MEC also improves generalization across different data distributions.  We perform transfer learning experiments and elaborate the results below.

---

> > ### Author Response · Authors · 2022-08-02
> > **Response to Reviewer QNyH (Part 2)**
> >
> > > **2. "Could you test more transfer learning experiments with fine-grained classification benchmarks?"**
> >
> > Thank you for the very helpful suggestion. We have already conducted the suggested transfer learning experiments on fine-grained classification benchmarks in supplementary material. Specifically, we perform linear probing and fine-tuning experiments on 11 fine-grained classification datasets, following the setup in SimCLR [A] and BYOL [B] papers. Please refer to Appendix D (L660-667) for more details about the experiments. And we also list the results in the table below.
> >
> > | Method          | Food101  | CIFAR10  | CIFAR100 | SUN397   | Cars     | Aircraft | VOC2007  | DTD      | Pets     | Caltech-101 | Flowers  |
> > |-----------------|----------|----------|----------|----------|----------|----------|----------|----------|----------|-------------|----------|
> > | Linear probing: |          |          |          |          |          |          |          |          |          |             |          |
> > | SimCLR [A]      | 68.4     | 90.6     | 71.6     | 58.8     | 50.3     | 50.3     | 80.5     | 74.5     | 83.6     | 90.3        | 91.2     |
> > | BYOL [B]        | 75.3     | 91.3     | **78.4** | 62.2     | **67.8** | 60.6     | 82.5     | 75.5     | 90.4     | 94.2        | **96.1** |
> > | MEC             | **75.6** | **92.1** | **78.4** | **62.7** | 67.2     | **61.5** | **82.7** | **75.8** | **90.9** | **94.6**    | 96.0     |
> > | Fine-tuned:     |          |          |          |          |          |          |          |          |          |             |          |
> > | Random init [A] | 86.9     | 95.9     | 80.2     | 53.6     | 91.4     | 85.9     | 67.3     | 64.8     | 81.5     | 72.6        | 92.0     |
> > | SimCLR [A]      | 88.2     | 97.7     | 85.9     | 63.5     | 91.3     | 88.1     | 84.1     | 73.2     | 89.2     | 92.1        | 97.0     |
> > | BYOL [B]        | 88.5     | **97.8** | 86.1     | 63.7     | **91.6** | 88.1     | 85.4     | **76.2** | 91.7     | 93.8        | 97.0     |
> > | MEC             | **88.9** | **97.8** | **86.8** | **63.8** | **91.6** | **88.5** | **85.9** | 76.0     | **91.9** | **94.9**    | **97.2** |
> >
> > The results show that the learned representations of MEC are more generalizable and less biased across the various data domains, compared to other models pre-trained with specific pretext tasks. We will add the results and discussions to the main paper to further demonstrate the transferability of the learned representations.
> >
> > > **3. "Too small top margins in p2-3." "Also, other spaces (e.g., caption spaces) seem too narrow."**
> >
> > Thank you for pointing these out. We have modified the top margins and other spaces accordingly in the updated version of the paper.
> >
> > **References**
> >
> > [A] Chen, Ting, et al. "A simple framework for contrastive learning of visual representations." ICML 2020.
> >
> > [B] Grill, Jean-Bastien, et al. "Bootstrap your own latent-a new approach to self-supervised learning." NeurIPS 2020.
> >
> > [C] Xie, Enze, et al. "Detco: Unsupervised contrastive learning for object detection." CVPR 2021.
> >
> > [D] Zbontar, Jure, et al. "Barlow twins: Self-supervised learning via redundancy reduction." ICML 2021.

---

> ### Comment · Reviewer_QNyH · 2022-08-05
> **Post Rebuttal Comment**
>
> Thank you to the authors for their response to the reviews.
>
> > we explicitly encourage the generalization ability on downstream tasks and minimize the bias in the formulation of the pretext task, by introducing the Maximum Entropy Principle.
>
> I still feel that the explanation about bias and learned information is unclear. First, the provided results can show MEC's superiority, but cannot explain how MEC can achieve the superiority. One can consider Maximum Entropy Principle as another pretext task. Moreover, MEC is also an image-level SSL objective relying on strong data augmentations. Therefore, MEC also has some bias introduced by the principle, image-level objective, and data augmentations. Can you guarantee MEC always has better bias than existing methods? Since this work introduces a new objective, I think it would be better to analyze the proposed method systemically, e.g., by addressing the following questions:
> - Which information (e.g., color, position, background, etc.) can be easier to learn from MEC than existing methods? This may be answered by checking mutual information between target information and the learned representation. (For example, see [this paper](https://openreview.net/forum?id=U34rQjnImpM), Figure 2.)
> - Why can MEC learn more patch/pixel-level information compared to other image-level SSL objectives?
>
> Nevertheless, I think the experimental results (including transfer learning for fine-grained classification tasks) are very strong, so I'm still positive about this paper. Hence, I keep my positive rating.

---

> > ### Author Response · Authors · 2022-08-08
> > **Thanks for the additional feedback**
> >
> > We sincerely thank the reviewer for the additional feedback and valuable comments.
> >
> > The maximum entropy principle states that the probability distribution that best represents the current state of knowledge about a system is the one with largest entropy, given a testable information, and in this way, no additional bias or assumptions is introduced. Our proposed method is based on the principle, which theoretically guarantees that the learned representations are less biased. And this is also supported by the empirical evidence on various downstream tasks.
> >
> > We are particularly inspired by the reviewer's comments that more empirical evidence can be leveraged to explicitly show what information the proposed method can learn and is beneficial for the downstream tasks. For example, following the method in [A], we can measure the mutual information between the learned representations and target information (e.g., color, position, patch/pixel), which will provide a more systematic analysis and make our method more interpretable. As noted by the reviewer, the bias can also be introduced by the designed data augmentations. To address this problem, we can further incorporate other methods that focus on this aspect (e.g., AugSelf [A]) into MEC to minimize the bias introduced by both data augmentations and pretext tasks.
> >
> > Thanks again for the very helpful suggestions which will be incorporated to make the paper stronger.
> >
> > **References**
> >
> > [A] Lee, Hankook, et al. "Improving transferability of representations via augmentation-aware self-supervision." NeurIPS 2021.

---

### Public Comment · ~Bizhe_Bai1 · 2023-02-28
**Wonderful work. Question about math prove**

Hello.Thanks for your wonderful work.
I have a question about section 2.2. In section 2.2, you mentioned, "Then they are fed to an encoder f,
consisting of a backbone and a projector network, which produces ℓ2-normalized embeddings of z1
and z2" .Is there any theoretical proof of producing ℓ2-normalized embeddings or any reference?
Thanks!

---

### Meta-Review · Area_Chair_RUus · 2022-08-24

**Recommendation:** Accept
**Confidence:** Certain

**Metareview:**

The paper in general received three positive feedbacks and ratings. The three reviewers all recognize the theoretical soundness of the paper, and the paper is also clearly presented with informative and strong experimental results. There is a few places making one reviewer less comfortable in terms of the exact effectiveness of the proposed theory. While overall the experimental results are comprehensive and basically suppor the claiming points made by the paper. The authors may furher clarify the points based on the comments.

**Award:**

No

---

### Decision · Program_Chairs · 2022-09-14

Accept